# MMAR: A Challenging Benchmark for Deep Reasoning in Speech, Audio, Music, and Their Mix

[1,2]**Ziyang Ma***[†], [3]**Yinghao Ma**,* [1]**Yanqiao Zhu**,* [1]**Chen Yang**,* [2]**Yi-Wen Chao**,* [1]**Ruiyang Xu**,*
[1]**Wenxi Chen**, [4]**Yuanzhe Chen**, [4]**Zhuo Chen**, [4]**Jian Cong**, [6]**Kai Li**, [7]**Keliang Li**, [3]**Siyou Li**,
[2]**Xinfeng Li**, [1]**Xiquan Li**, [7]**Zheng Lian**, [1]**Yuzhe Liang**, [8]**Minghao Liu**, [1,5]**Zhikang Niu**,
[2]**Tianrui Wang**, [4]**Yuping Wang**, [4]**Yuxuan Wang**, [2]**Yihao Wu**, [1]**Guanrou Yang**,
**Jianwei Yu**, [9]**Ruibin Yuan**, [10]**Zhisheng Zheng**, [9]**Ziya Zhou**, [1]**Haina Zhu**,
[9]**Wei Xue**, [3]**Emmanouil Benetos**, [1]**Kai Yu**, [2]**Eng-Siong Chng**, [1,5]**Xie Chen**[‡]

[1]Shanghai Jiao Tong University, [2]Nanyang Technological University,
[3]Queen Mary University of London, [4]ByteDance, [5]Shanghai Innovation Institute,
[6]Tsinghua University, [7]University of Chinese Academy of Sciences, [8]2077AI,
[9]The Hong Kong University of Science and Technology, [10]The University of Texas at Austin

`https://github.com/ddlBoJack/MMAR`

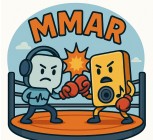

## Abstract

We introduce MMAR, a new benchmark designed to evaluate the deep reasoning capabilities of Audio-Language Models (ALMs) across massive multi-disciplinary tasks. MMAR comprises 1,000 meticulously curated audio-question-answer triplets, collected from real-world internet videos and refined through iterative error corrections and quality checks to ensure high quality. Unlike existing benchmarks that are limited to specific domains of sound, music, or speech, MMAR extends them to a broad spectrum of real-world audio scenarios, including mixed-modality combinations of sound, music, and speech. Each question in MMAR is hierarchically categorized across four reasoning layers: Signal, Perception, Semantic, and Cultural, with additional sub-categories within each layer to reflect task diversity and complexity. To further foster research in this area, we annotate every question with a Chain-of-Thought (CoT) rationale to promote future advancements in audio reasoning. Each item in the benchmark demands multi-step deep reasoning beyond surface-level understanding. Moreover, a part of the questions requires graduate-level perceptual and domain-specific knowledge, elevating the benchmark's difficulty and depth. We evaluate MMAR using a broad set of models, including Large Audio-Language Models (LALMs), Large Audio Reasoning Models (LARMs), Omni Language Models (OLMs), Large Language Models (LLMs), and Large Reasoning Models (LRMs), with audio caption inputs. The performance of these models on MMAR highlights the benchmark's challenging nature, and our analysis further reveals critical limitations of understanding and reasoning capabilities among current models. These findings underscore the urgent need for greater research attention in audio-language reasoning, including both data and algorithm innovation. We hope MMAR will serve as a catalyst for future advances in this important but little-explored area.

---

*Core Contributors. Other authors are listed in surname alphabetical order.
[†]Project Leader.
[‡]Corresponding Author.

39th Conference on Neural Information Processing Systems (NeurIPS 2025) Track on Datasets and Benchmarks.

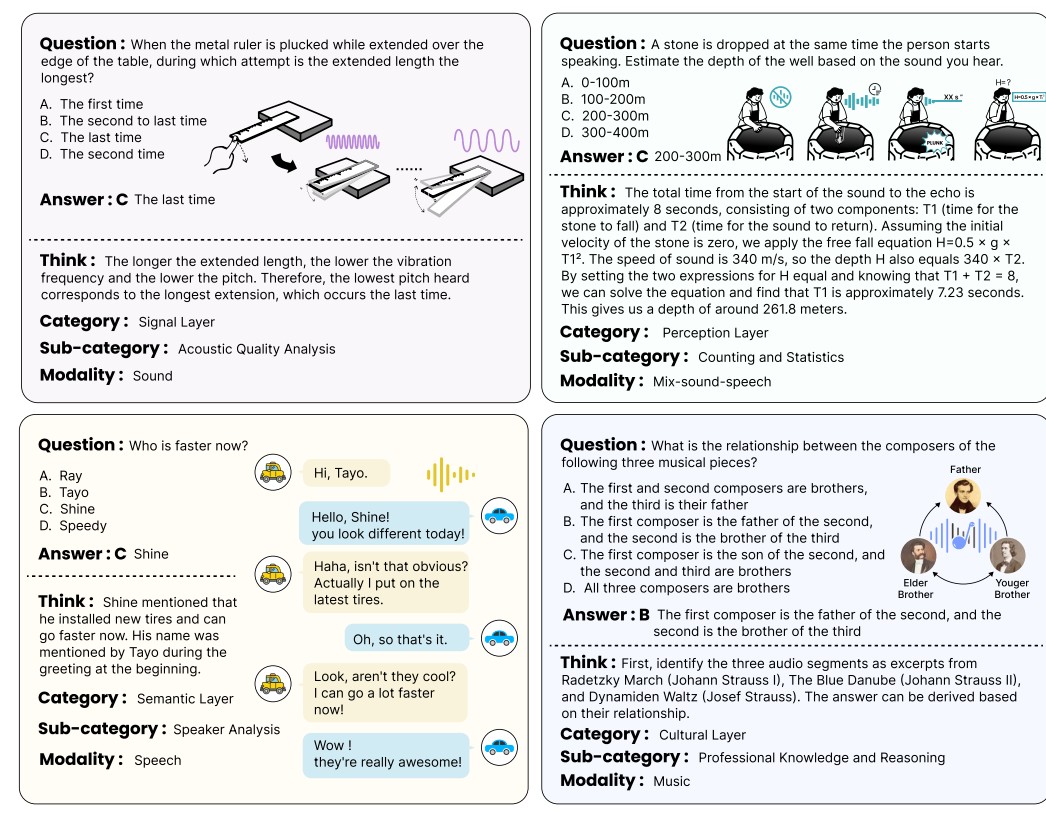

Figure 1: Examples from the MMAR benchmark, illustrating challenges at the signal, perceptual, semantic, and cultural levels. The examples span audio, speech, music, and their mix.

# 1 Introduction

With the rapid advancements in large language models (LLMs) and audio processing technologies, large audio language models (LALMs) [17, 9, 10, 7, 2, 3, 30] have emerged as a powerful paradigm that combines an audio encoder for acoustic signal processing with an LLM for text processing. These models have demonstrated impressive performance across a wide range of tasks, including automatic speech recognition [38, 26], audio captioning [22, 1], and music analysis [4, 23]. Significantly enhancing machine auditory capabilities, LALMs represent a critical step toward embodied intelligence and the broader goal of artificial general intelligence (AGI). Recently, models such as OpenAI o1 [16] and DeepSeek-R1 [13] have shown that scaling inference can substantially improve reasoning ability. Inspired by this, a new wave of large audio reasoning models (LARMs) has been proposed, aiming to tackle more complex reasoning tasks grounded in audio through prompt engineering [25], supervised fine-tuning [35], or reinforcement learning [18, 34]. However, the audio domain still lacks a rigorous deep reasoning benchmark analogous to MMLU-Pro [32] in the textual domain. This absence poses a significant barrier to progress in evaluating and advancing audio deep reasoning capabilities.

To address this gap, we introduce MMAR, a new benchmark designed to evaluate the deep reasoning[4] capabilities of Audio-Language Models (ALMs) across a diverse set of multi-disciplinary tasks. As shown in Figure 1, MMAR includes questions that require multi-step reasoning over different types of audio inputs, such as sound, speech, music, and mixed-modality (e.g., mix-sound-speech). We define a hierarchical taxonomy of tasks: Signal, Perception, Semantic, and Cultural layers, which is co-developed through Human–LLM collaboration. Each question is also annotated with fine-grained sub-categories and a manually labeled Chain-of-Thought (CoT), which explicitly traces the reasoning process and supports future research. The construction of the MMAR benchmark follows a detailed

---

[4]We define *audio deep reasoning* as tasks that require expert-level perceptual understanding, multi-step logical inference, and the application of contextual or domain-specific knowledge to interpret complex audio inputs. These tasks are often challenging even for humans, typically demanding deliberate reasoning or specialized expertise.

pipeline involving expert question authorship, multi-stage refinement, and rigorous quality control, ensuring high-quality and reliable annotations.

Based on the MMAR benchmark, we evaluate 30 audio-capable models, including 24 open-source and 6 closed-source models, spanning LALMs, LARMs, Omni Language Models (OLMs), LLMs, and Large Reasoning Models (LRMs) with audio caption inputs. Our analysis yields several key findings: (1) The MMAR benchmark is highly challenging. None of the evaluated open-source LALMs perform significantly better than random guessing, highlighting the substantial difficulty in MMAR. (2) There exists a notable performance gap between open-source and closed-source models. Among open-source models, Qwen-2.5-Omni [36] achieves the best results. However, Gemini 2.0 Flash [12] outperforms all models, including cascaded reasoning pipelines, underscoring the advantage of tightly integrated closed-source systems. (3) Across both end-to-end and cascaded settings, reasoning-enhanced models consistently outperform non-reasoning models. This demonstrates the critical role of explicit reasoning mechanisms in handling MMAR's deep reasoning tasks. These insights point to an urgent need for the open-source community to develop stronger LARMs.

In summary, our contributions are as follows:

1. We present MMAR, the first benchmark specifically designed to evaluate deep reasoning in the audio domain. MMAR features high-quality human annotations, mixed-modality coverage, hierarchical task taxonomy, and different reasoning questions, making it a uniquely comprehensive benchmark.
2. We benchmark 30 audio-capable models across five model categories, revealing that current open-source models struggle significantly with audio deep reasoning.
3. We conduct extensive analysis and comparison experiments, identifying key challenges and architectural limitations. Our findings provide valuable insights for developing next-generation LARMs.

## 2 Related Work

### 2.1 Audio-Language Models

Recent advancements in multimodal AI have led to the development of a variety of models that accept audio as input and perform diverse downstream tasks. These models generally fall into three major categories: (1) Large Audio-Language Models (LALMs), (2) Large Audio Reasoning Models (LARMs), and (3) Omni Language Models (OLMs).

LALMs combine audio encoders with large language models to enable joint audio-text understanding. Some models, such as Audio Flamingo [17], Audio Flamingo 2 [9], LTU-AS [10], and Qwen-Audio [2], use a single encoder for all audio types. Other models, like SALMONN [30] employing multiple encoders or GAMA [7] with several specialized projectors, better handle diverse audio domains. To further enhance performance, Qwen2-Audio [3] applies reinforcement learning with human feedback (RLHF) for better alignment with human reasoning. To tackle more complex inference tasks, LARMs build on LALMs by incorporating explicit reasoning mechanisms. Representative models such as Audio-CoT [25], Audio-Reasoner [35], and R1-AQA [18] are developed by augmenting Qwen2-Audio-Instruct [3]. SARI [34], built upon by Qwen-2.5-Omni [36], performs better on audio reasoning through a thinking process. OLMs are general-purpose multimodal models designed to handle both multimodal input and output. While not specifically designed for audio, they show strong generalization ability across modalities. Examples include open-source models like AnyGPT [39], OpenOmni [24], Baichuan-Omni [19], and Qwen-2.5-Omni [36], as well as closed-source systems like Gemini 2.0 Flash [12].

While prior benchmarks have primarily focused on evaluating only LALMs, our work goes further by benchmarking all three model types, also including cascaded systems with LLMs and LRMs that convert audio into text before reasoning.

### 2.2 Audio Understanding & Reasoning Benchmarks

Several benchmarks have been proposed to evaluate the understanding capabilities of LALMs, such as Clotho-AQA [20] and CompA [8] for sound, MusicBench [27] and MuChoMusic [33] for music, and LibriSQA [40] and Dynamic-SUPERB [14] for speech. Broader benchmarks such as AudioBench [31], AIR-Bench [37], and MMAU [29] combine multiple audio domains. However, these benchmarks primarily assess surface-level understanding tasks, with limited evaluation of reasoning capabilities, particularly deep reasoning, which is the central focus of MMAR.

While audio deep reasoning is a critical yet underexplored component, what sets MMAR apart is its comprehensive, multi-dimensional design. As illustrated in Table 1, MMAR offers several key advantages over existing benchmarks:

**Real-world Mixed-modality Audio Coverage.** Unlike prior benchmarks such as AudioBench [31] and MMAU [29], which focus only on unimodal domains (i.e., sound, music, or speech), MMAR includes naturally occurring mixed-modality audio. Although AIR-Bench [37] attempts to simulate it, its audio is artificially synthesized by combining unimodal clips. In contrast, MMAR features real-world audio scenarios where complex interactions among sound, speech, and music are inherent. Some tasks even require models to reason jointly across all three modalities.

**Explicit Evaluation of Deep Reasoning.** While benchmarks like MMAU [29] include some reasoning tasks, these are typically shallow in complexity. AIR-Bench [37] introduces LLM-generated reasoning problems, but these lack human-level scrutiny and depth. Every question in MMAR involves multi-step reasoning, and some require graduate-level perceptual and domain-specific knowledge. This makes MMAR the first benchmark to explicitly focus on deep audio reasoning.

**Newly Curated Data to Prevent Leakage.** Many existing benchmarks reuse data from well-known datasets such as AudioSet [6], which may have been seen by pre-trained models, especially in LALM settings. In contrast, all audio data in MMAR is newly collected from online videos, ensuring both diversity and no data leakage, thus providing a more reliable and future-proof evaluation.

In summary, MMAR not only advances the field by introducing mixed modalities, real-world audio and deep reasoning challenges, but also sets a higher standard for data quality and originality in benchmark construction.

Table 1: Comparison of audio-language model benchmarks across four key dimensions: (i) **Domain Coverage**—whether the benchmark spans diverse audio types such as speech, sound events, music, or mixtures; (ii) **Task Scope**—ranging from basic understanding to deep reasoning; (iii) **Evaluation Paradigm**—whether the benchmark treats each task as a traditional task-specific evaluation or as a sample-level assessment (**Sample-As-A-Task**); and (iv) **Data Origin**—whether the benchmark is newly collected or sourced from existing datasets. ✔ indicates audio data or question-answer pairs that are artificially synthesized, rather than in-the-wild or handcrafted by experts.

| Benchmark | Domain | | | | Scope | | | | Sample-As-A-Task | Newly Collected |
| | Sound | Music | Speech | Mix | Understanding | Graduate-Level Understanding | Reasoning | Deep Reasoning | | |
|---|---|---|---|---|---|---|---|---|---|---|
| **AudioBench [31]** | ✔ | ✔ | ✗ | ✗ | ✔ | ✗ | ✗ | ✗ | ✗ | ✗ |
| **AIR-Bench [37]** | ✔ | ✔ | ✔ | ✔ | ✔ | ✗ | ✔ | ✗ | ✗ | ✗ |
| **MMAU [29]** | ✔ | ✔ | ✔ | ✗ | ✔ | ✗ | ✔ | ✗ | ✔ | ✗ |
| **MMAR** (ours) | ✔ | ✔ | ✔ | ✔ | ✔ | ✔ | ✔ | ✔ | ✔ | ✔ |

# 3 The MMAR Benchmark

## 3.1 Overview

MMAR is a benchmark designed to evaluate the reasoning capabilities of Audio-Language Models (ALMs). It consists of 1,000 meticulously handcrafted audio deep reasoning tasks, each requiring multi-step inference, and some of these tasks demand highly challenging perceptual skills and domain-specific knowledge. The questions were developed by domain experts and subsequently refined and validated through expert review and quality assurance to ensure correctness and high quality. Figure 2 illustrates MMAR's data distribution along two key dimensions: modality coverage and task coverage, as well as audio and metadata statistics.

**Domain Coverage.** Unlike prior benchmarks that focus on unimodal audio types, such as sound, music, and speech, MMAR incorporates a broader set of real-world, mixed-modality audio scenarios. This reflects the reality that natural audio environments often contain complex combinations of sub-modalities. As shown in Figure 2a, MMAR includes seven distinct audio domain categories: sound, music, speech, mix-sound-music, mix-sound-speech, mix-music-speech, and mix-sound-music-speech.

**Task Coverage.** Leveraging expert brainstorming and Human–LLM collaboration, we constructed a four-level hierarchical reasoning taxonomy ranging from concrete to abstract, as shown in Figure 2b. These levels include the signal layer, perception layer, semantic layer, and cultural layer. Each layer

is further subdivided into multiple subcategories, representing a wide variety of reasoning tasks. Definitions and examples for each reasoning layer are provided in Appendix A.

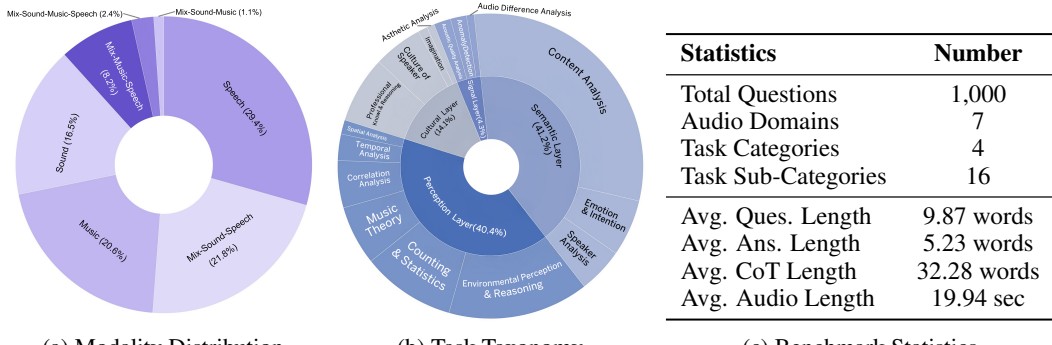

| Statistics | Number |
|---|---|
| Total Questions | 1,000 |
| Audio Domains | 7 |
| Task Categories | 4 |
| Task Sub-Categories | 16 |
| Avg. Ques. Length | 9.87 words |
| Avg. Ans. Length | 5.23 words |
| Avg. CoT Length | 32.28 words |
| Avg. Audio Length | 19.94 sec |

|  (a) Modality Distribution | (b) Task Taxonomy | (c) Benchmark Statistics |

Figure 2: **(a)** The data distribution of single and mixed modalities in MMAR. **(b)** The hierarchical taxonomy and data distribution of task categories in MMAR. **(c)** Data statistics of the MMAR benchmark.

Table 2c presents key statistics of the MMAR benchmark. During the annotation process, expert question designers were trained to formulate questions with high precision, provide concise answers, and construct detailed chains of thought. The resulting average lengths of Question, Answer, and CoT components are summarized in the table. In addition, considering that most current models require input audio to be under 30 seconds, we instructed question designers to adhere to this constraint. As a result, the average audio length in MMAR is approximately 20 seconds. In comparison, the average audio duration in the previous MMAU [29] benchmark was around 10 seconds, suggesting that MMAR audio clips contain significantly richer and more complex information content.

## 3.2 Data Curation Pipeline

As shown in Figure 3, we constructed the MMAR benchmark through a five-stage pipeline:

**1) Brainstorming.** Given that deep reasoning on audio is a novel and complex task, which requires sophisticated fusion of acoustic and linguistic information, high-quality question ideation is particularly challenging. To address this, we organized multiple rounds of brainstorming sessions with expert annotators, collecting a wide range of unstructured cognitive fragments and reasoning sketches. The qualifications of the expert annotators involved can be found in Appendix B.

**2) Taxonomy Construction.** We employed the LLM to extract and organize insights from the brainstorming sessions, collaborating with experts to build a hierarchical taxonomy ranging from abstract to concrete task levels. An initial set of sub-categories was also established during this phase.

**3) Heuristic-Based Human Annotation.** Leveraging the established taxonomy and brainstorm outputs, annotators heuristically searched for relevant internet videos and manually labeled each data instance. Annotations included: video URL and timestamps, question, answer, chain of thought, audio modality, task category and sub-category, and spoken language (if present).

**4) Raw Data Preparation.** Based on the annotated metadata, we proceeded along two parallel paths: (1) Audio data collection, which involved crawling and trimming audio clips for each question and performing additional processing for complex cases (e.g., comparisons across clips, audio reversal, etc.); (2) LLM-based content generation, where we refined and enhanced the original CoTs and generated distractor options for multiple-choice questions. Most questions have four options, while a minority (e.g., binary or ternary questions) have two or three. We also refined the task sub-categories based on question content to ensure coherence. This stage produced a raw, unverified JSON file along with the corresponding audio data.

**5) Data Quality Inspection.** Ensuring high data quality was central to the construction of MMAR benchmark. We used a dedicated professional data annotation platform as shown in Appendix C, and engaged domain experts for correction and quality inspection. We enforced a dual guarantee of quality: (1) Separation of roles: Each question was independently authored, corrected, and reviewed by different individuals; (2) Iterative revision: Any question that failed more than two rounds of inspection was discarded. Ultimately, 1,000 high-quality questions were selected for inclusion in the final MMAR benchmark.

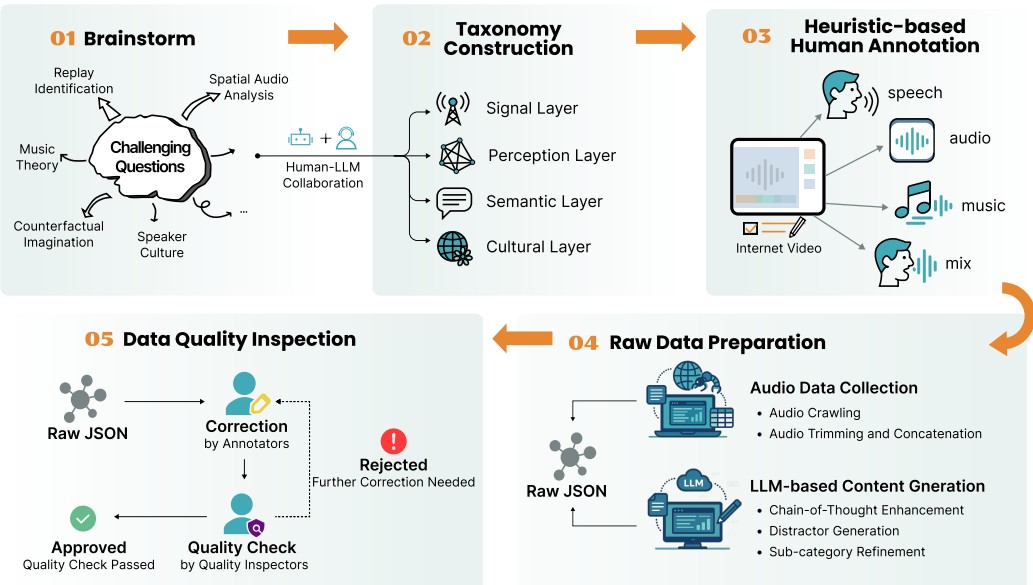

Figure 3: A comprehensive pipeline for constructing the MMAR benchmark. The process includes: (1) brainstorming challenging questions; (2) building a taxonomy through human-LLM collaboration; (3) heuristic-based data collection and annotation; (4) crawling audio data and enriching metadata content; and (5) performing iterative correction and quality inspection to ensure high data fidelity.

## 4 Experimental Setup

### 4.1 Benchmarking Candidates

We evaluate five categories of audio-capable models: (1) Large Audio Language Models (LALMs), designed for audio-text understanding; (2) Large Audio Reasoning Models (LARMs), which enhance LALMs with explicit reasoning chains; (3) Omni Language Models (OLMs), supporting fully multimodal input/output; (4) Large Language Models (LLMs) with audio captions, which process Qwen2-Audio-Instruct-generated captions; and (5) Large Reasoning Models (LRMs), which perform reasoning over captions using models with inference scaling. Further details and model configurations are provided in Appendix E.

### 4.2 Evaluation Methods

Since all MMAR tasks are formulated as multiple-choice questions, we adopt classification accuracy as the evaluation metric. Specifically, we input the audio, question, and choices into each model and evaluate whether the model selects the correct option. To determine correctness, we follow the same approach as MMAU [29], using regular expressions and string matching to compare the model's prediction with ground truth answers. For models without explicit reasoning, we directly evaluate the model's final prediction. For models with explicit reasoning chains, we remove the thinking content and evaluate only the final predicted answer to ensure fairness and consistency across model types.

## 5 Experimental Results

The results presented in Table 2 offer several key insights into the difficulty of MMAR and the current capabilities of audio-language models. All questions are multiple-choice tests with a variable number of options. Models are evaluated on seven domains spanning single and mixed audio modalities, with accuracy (%) reported.

**First, the overall performance across all model categories confirms that MMAR is a highly challenging benchmark.** As shown in Table 2, even the strongest open-source model, Qwen-2.5-Omni (7B), achieves an average accuracy below 60%. Moreover, when analyzing performance across different modalities, we observe that music-related tasks are particularly challenging, with significantly lower accuracy compared to other modalities. Figure 4 visualizes statistical significance using the Poisson Binomial distribution, highlighting whether model predictions are significantly better than random guessing. We apply the Bonferroni correction (see at Appendix M) to adjust the p-value threshold for multiple comparisons. As the figure shows, none of the open-source LALMs

Table 2: MMAR results across five model categories: LALMs, LARMs, OLMs, LLMs, and LRMs with audio captions as input. The best-performing models in each category are highlighted in **bold**, and the second-best ones are underlined.

| Models | Size | Single Modality (%) | | | Mixed Modalities (%) | | | | Avg (%) |
| --- | --- | --- | --- | --- | --- | --- | --- | --- | --- |
| | | Sound | Music | Speech | Sound-Music | Sound-Speech | Music-Speech | Sound-Music-Speech | |
| Random Guess | - | 29.39 | 25.88 | 31.48 | 25.00 | 29.30 | 31.10 | 28.13 | 29.32 |
| Large Audio Language Models (LALMs) | | | | | | | | | |
| Audio Flamingo [17] | 2.2B | 32.73 | 21.84 | 24.83 | 18.18 | 30.28 | 24.39 | 25.00 | 26.60 |
| Audio Flamingo 2 [9] | 0.5B | 20.61 | 20.39 | 24.15 | 27.27 | 23.85 | 26.83 | 25.00 | 23.00 |
| Audio Flamingo 2 [9] | 1.5B | 26.67 | 20.87 | 22.79 | 9.09 | 22.94 | 23.17 | 20.83 | 22.90 |
| Audio Flamingo 2 [9] | 3B | 24.85 | 17.48 | 20.75 | 18.18 | 26.61 | 23.17 | 8.33 | 21.90 |
| LTU [11] | 7B | 19.39 | 19.90 | 13.95 | 18.18 | 24.77 | 21.95 | 16.67 | 19.20 |
| LTU-AS [10] | 7B | 20.00 | 14.08 | 19.05 | 9.09 | 20.64 | 28.05 | 12.50 | 19.00 |
| MusiLingo [4] | 7B | 9.09 | 7.28 | 4.08 | 9.09 | 6.88 | 7.32 | 8.33 | 6.60 |
| MU-LLaMA [23] | 7B | 13.94 | 13.59 | 14.97 | 9.09 | 12.39 | 14.63 | 16.67 | 13.90 |
| GAMA [7] | 7B | 29.09 | 24.27 | 27.89 | 27.27 | 24.77 | 28.05 | 20.83 | 26.50 |
| GAMA-IT [7] | 7B | 22.42 | 16.02 | 12.24 | 36.36 | 22.48 | 14.63 | 12.50 | 17.40 |
| Qwen-Audio-Chat [2] | 8.4B | 27.88 | 20.39 | 22.11 | 9.09 | 25.23 | 25.61 | 20.83 | 23.50 |
| Qwen2-Audio [3] | 8.4B | 33.94 | 23.30 | 32.99 | 9.09 | 33.03 | 26.83 | 33.33 | 30.40 |
| Qwen2-Audio-Instruct [3] | 8.4B | 33.33 | 24.27 | 32.31 | 9.09 | 31.19 | 30.49 | 25.00 | 30.00 |
| SALMONN [30] | 7B | 30.91 | 29.61 | 34.35 | 9.09 | 37.61 | 28.05 | 37.50 | 32.80 |
| SALMONN [30] | 13B | 30.30 | 31.07 | 34.69 | 9.09 | 34.86 | 35.37 | 41.67 | 33.20 |
| GPT-4o mini Audio [15] | - | 38.79 | 35.92 | 58.84 | 45.45 | 60.09 | 57.32 | 50.00 | 50.60 |
| GPT-4o Audio [15] | - | 53.94 | **50.97** | 70.41 | 63.64 | **72.48** | 62.20 | **75.00** | 63.50 |
| Large Audio Reasoning Models (LARMs) | | | | | | | | | |
| Mellow [5] | 167M | 33.33 | 26.70 | 24.83 | 18.18 | 37.16 | 32.93 | 29.17 | 30.00 |
| Audio-CoT [25] | 8.4B | 35.76 | 25.24 | 34.01 | 9.09 | 30.73 | 30.49 | 37.50 | 31.30 |
| Audio-Reasoner [35] | 8.4B | 43.64 | 33.50 | 32.99 | 45.45 | 42.66 | 31.71 | 25.00 | 36.80 |
| Omni Language Models (OLMs) | | | | | | | | | |
| AnyGPT-chat [39] | 8B | 24.24 | 19.42 | 22.11 | 27.27 | 27.52 | 26.83 | 29.17 | 23.70 |
| OpenOmni [24] | 8B | 20.61 | 22.33 | 35.37 | 18.18 | 27.06 | 23.17 | 25.00 | 27.00 |
| Baichuan-Omni-1.5 [19] | 11B | 41.21 | 33.01 | 40.48 | 36.36 | 48.62 | 39.02 | 41.67 | 40.70 |
| Qwen-2.5-Omni [36] | 3B | 53.94 | 46.12 | 53.74 | 36.36 | 60.09 | 57.32 | 58.33 | 53.80 |
| Qwen-2.5-Omni [36] | 7B | 58.79 | 40.78 | 59.86 | 54.55 | 61.93 | **67.07** | 58.33 | 56.70 |
| Gemini 2.0 Flash [12] | - | **61.21** | **50.97** | **72.11** | **81.82** | **72.48** | 65.85 | 70.83 | **65.60** |
| Large Language Models (LLMs) | | | | | | | | | |
| Caption + DeepSeek-V3 [21] | 671B | 42.42 | 40.78 | 56.12 | 18.18 | 50.00 | 45.12 | 37.50 | 47.60 |
| Caption + GPT-4o [15] | - | 46.06 | 40.29 | 60.88 | 27.27 | 53.67 | 46.34 | 45.83 | 50.70 |
| Large Reasoning Models (LRMs) | | | | | | | | | |
| Caption + DeepSeek-R1 [13] | 671B | 46.67 | 49.51 | 62.59 | 45.45 | 58.72 | 56.10 | 54.17 | 55.50 |
| Caption + OpenAI o1 [16] | - | 48.48 | 43.20 | 63.61 | 18.18 | 56.88 | 45.12 | 45.83 | 53.00 |
| Caption + OpenAI o3 [28] | - | 49.70 | 41.75 | 63.95 | 36.36 | 60.09 | 52.44 | 54.17 | 54.70 |

achieve statistically significant improvements over random guessing. Additionally, in Appendix F, we compare several competitive models on both MMAU and MMAR, showing that MMAR is substantially more difficult across the board. This highlights MMAR's emphasis on multi-step reasoning and rich audio content, placing it well beyond the scope of traditional audio question answering benchmarks.

**Second, there exists a significant performance gap between open-source and closed-source models.** Among open-source models, Qwen-2.5-Omni (7B) achieves the highest accuracy at 56.7%, clearly outperforming other LALMs and LARMs. However, it still lags behind the best-performing closed-source model, Gemini 2.0 Flash, which reaches an impressive 65.6%. Notably, Gemini 2.0 Flash outperforms all cascaded setups (e.g., captioning followed by LLM or LRM), demonstrating the effectiveness of well-integrated multimodal architectures.

**Third, we observe that models with explicit reasoning capabilities consistently outperform those without, regardless of whether the architec-**

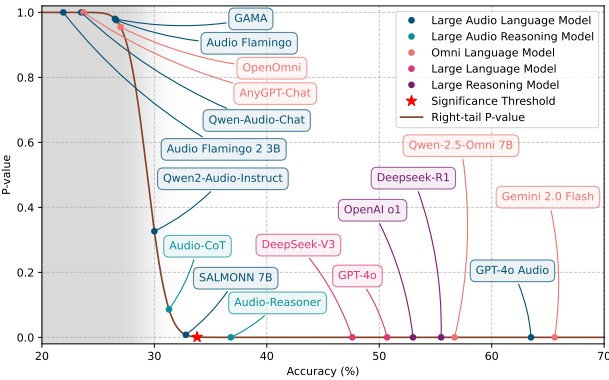

Figure 4: Poisson Binomial distribution illustrating the P-Value of accuracy under random guessing. For LLMs and LRMs, audio captions are provided. One-tailed right-sided P-value is computed to test whether each model significantly outperforms random guessing. Solid circles mark the observed performance of each model. A red star indicates the Bonferroni-corrected significance threshold ($\alpha = 0.001$).

**ture is end-to-end or cascaded.** For example, Audio-Reasoner surpasses Qwen2-Audio and Qwen2-Audio-Instruct, and Caption + DeepSeek-R1 outperforms Caption + DeepSeek-V3. This trend suggests that training reasoning models is crucial for handling the challenges posed by MMAR, and reasoning-enhanced architectures are better equipped to catch complex multimodal interactions.

Overall, these findings underline the need for further innovation in audio-language reasoning, particularly in open-source research, where the performance gap remains substantial.

## 6 Discussion

### 6.1 Comparison on Task Hierarchies

Figure 5a presents a comparison of model performance across the four hierarchical task layers in MMAR, ordered from concrete to abstract. For visualization, we select one representative model from each category: a LALM (Qwen2-Audio-Instruct), a LARM (Audio-Reasoner), and an OLM (Qwen-2.5-Omni). The gray bars represent random guess baselines. From the figure, we observe that all models perform best on the Semantic Layer, while achieving the lowest accuracy on the Signal Layer, despite the fact that random guessing performs highest on Signal Layer. A possible reasoning is that semantic tasks often involve understanding spoken language, where models benefit from abundant pretraining data. In contrast, signal-level tasks demand fine-grained reasoning about low-level physical properties of audio, which are less frequently encountered and underrepresented in model training data.

### 6.2 Comparison with Noise Input

Figure 5b shows model performance on MMAR when the original audio inputs are replaced with noise. This experiment serves two purposes: (1) to evaluate whether models are truly leveraging the audio input, and (2) to examine the role of language priors in ALMs. From the figure, we observe that even the weakest model, Qwen2-Audio-Instruct, experiences a substantial performance drop when audio is replaced with noise, indicating that all models are indeed listening to the audio rather than relying solely on textual or statistical biases. However, it is noteworthy that Qwen-2.5-Omni still performs slightly above random guessing even with noise input. This suggests a residual language prior bias, despite our three-stage quality control pipeline where question designers, reviewers, and inspectors were explicitly instructed to ensure that questions could not be answered from text alone. This finding highlights an important consideration for future multimodal benchmarks: even subtle text-based patterns can introduce unintended cues that powerful language models may exploit.

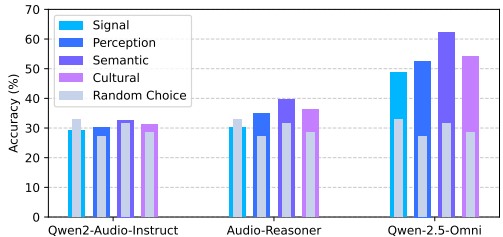

(a) Compare different reasoning hierarchies on MMAR

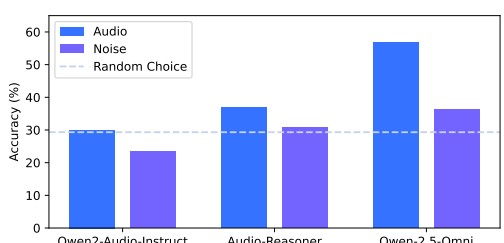

(b) Compare with noise input on MMAR

Figure 5: Performance comparison on the MMAR benchmark. **(a)** compares task accuracy on different reasoning hierarchies, and **(b)** shows the impact of using noise as input rather than audio.

### 6.3 Comparison on Cascaded Models

Table 3 presents a comparison of cascaded models built from different combinations of audio captioning and language models for downstream reasoning on MMAR. Specifically, we evaluate two audio captioning front-ends (Qwen2-Audio-Instruct and Qwen-2.5-Omni) paired with two powerful LLMs (GPT-4o and OpenAI o1). Results show that replacing either component leads to performance gains: swapping the captioning model from Qwen2-Audio-Instruct to Qwen-2.5-Omni improves accuracy, as does replacing the LLM from GPT-4o to o1. These findings suggest that both better auditory perception

Table 3: Compare audio caption models and LLMs in cascaded models.

| Reasoning\Perception | Qwen2-Audio-Instruct | Qwen-2.5-Omni |
|---|---|---|
| **GPT-4o** | 50.70 | 51.80 |
| **OpenAI o1** | 53.00 | 54.40 |

and stronger reasoning/knowledge abilities contribute independently and cumulatively to performance on MMAR, further validating the benchmark's sensitivity to both perception and reasoning capacity.

## 6.4 Error Analysis

To better understand the limitations of ALMs, we conduct a fine-grained error analysis on 100 failed predictions from Audio-Reasoner. Each case is manually categorized into four error types: perceptual errors, knowledge errors, reasoning errors, and other errors, given with examples in Table 4.

Table 4: Illustration of different error types on the well depth estimation question (the second example in Figure 1).

| Error Type | Incorrect Reasoning Path | Correct Reasoning Path |
|---|---|---|
| **Perceptual Error** (Misheard time: e.g., 6s instead of 8s) | Assume total time $T_1 + T_2 = 6$ s $\Rightarrow$ Use $H = 0.5gT_1^2$, $H = v_sT_2$ $\Rightarrow$ Solve with $T_1 + T_2 = 6$ $\Rightarrow T_1 \approx 5.5$ s, $H \approx 150$ m Answer: B (100-200m) | Correct time heard: $T_1 + T_2 = 8$ s $\Rightarrow$ Use $H = 0.5gT_1^2$, $H = v_sT_2$ $\Rightarrow$ Solve with $T_1 + T_2 = 6$ $\Rightarrow T_1 \approx 7.23$ s, $H \approx 261.8$ m Answer: C (200-300m) |
| **Knowledge Error** (Did not use sound return time) | Only use $H = 0.5gT^2$ with $T = 8$ s $\Rightarrow H = 0.5 \times 9.8 \times 8^2 = 313.6$ m Ignores $T_2$ (sound travel) Answer: D (300-400m) | Correctly use both falling time and sound return time $\Rightarrow$ Model total time as $T_1 + T_2$ $\Rightarrow$ Solve coupled equations Answer: C (200-300m) |
| **Reasoning Error** (Mistake in solving equation) | Use $T_1 + T_2 = 8$ $H = 0.5gT_1^2 = v_s(8 - T_1)$ Math error: Solve and get $T_1 = 6.5$ $\Rightarrow H = 0.5 \times 9.8 \times 6.5^2 \approx 206.3$ m Answer: B (100-200m) | Use $T_1 + T_2 = 8$ $H = 0.5gT_1^2 = v_s(8 - T_1)$ Correctly solve: $0.5gT_1^2 = v_s(8 - T_1)$ $\Rightarrow T_1 \approx 7.23$, $H \approx 261.8$ m Answer: C (200-300m) |

As summarized in Figure 6, the most prevalent issues stem from perceptual errors (37%), where the model struggles with core audio understanding tasks such as distinguishing environmental sounds, identifying musical structures, and accurately transcribing or interpreting speech. These errors often reflect the model's limited capacity to process detailed auditory context, resolve polyphonic overlaps, or perform fine-grained audio analysis. In addition, reasoning errors (20%) and knowledge gaps (9%) highlight the model's difficulty in multi-hop inference and domain grounding. Failures include misinterpreting causal structure, misunderstanding sarcasm, or lacking commonsense and cultural knowledge. Notably, a significant portion of errors (34%) fall into the "other" category, covering issues such as instruction misinterpretation, generation collapse, and misalignment between the

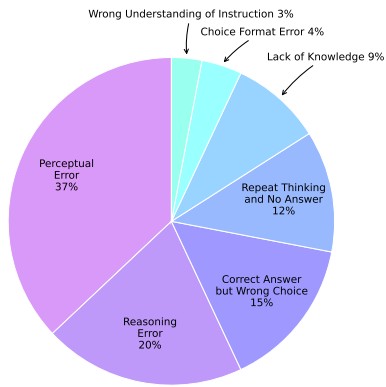

Figure 6: Error distribution in Audio-Reasoner.

reasoning chain and final choice. These findings reveal not only the complexity of auditory reasoning but also the fragility of current models in aligning perception, reasoning, and structured output.

## 7 Conclusion

In this work, we present MMAR, a new benchmark designed to evaluate the deep reasoning capabilities of Audio-Language Models (ALMs) across a broad range of real-world, mixed-modality, and multi-disciplinary tasks. MMAR comprises 1,000 high-quality, human-curated questions that span four hierarchical reasoning layers, each annotated with detailed chains of thought (CoT) to promote interpretability and future research. We benchmarked 30 audio-capable models across five categories, including LALMs, LARMs, OLMs, LLMs, and LRMs with caption input. Our results demonstrate that MMAR is substantially more challenging than existing benchmarks. Notably, open-source LALMs perform only marginally above random guessing, while reasoning-augmented and closed-sourced models show significantly stronger performance. Ablation studies and error analysis further reveal MMAR's usability and key limitations in current models. We hope MMAR will serve as a rigorous and forward-looking benchmark for advancing audio reasoning.

## Limitations

We acknowledge several limitations of the current benchmark. First, despite enforcing a three-stage annotation and review process requiring annotators to listen to the audio, ablation experiments reveal that a small subset of questions may still be answerable using text priors alone—indicating residual bias. Second, due to the high annotation difficulty (each item taking 10–30 minutes per round), the dataset size is limited to 1,000 examples. We recognize this as a constraint on coverage and plan to expand MMAR in the future, particularly in underrepresented sub-categories and modalities.

## Acknowledgements

This work was supported by the National Natural Science Foundation of China (No. 62206171 and No. U23B2018), Shanghai Municipal Science and Technology Major Project under Grant 2021SHZDZX0102 and Yangtze River Delta Science and Technology Innovation Community Joint Research Project (2024CSJGG01100).

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

# NeurIPS Paper Checklist

1. **Claims**

   Question: Do the main claims made in the abstract and introduction accurately reflect the paper's contributions and scope?

   Answer: [Yes]

   Justification: The main claims in the abstract and introduction are well-aligned with the paper's actual contributions and scope. Specifically, the authors introduce MMAR as a benchmark for evaluating deep reasoning in audio-language tasks across diverse modalities (speech, music, environmental sound) and reasoning levels. This claim is substantiated by detailed benchmark construction, multi-model evaluations, error analysis, and ablation studies throughout the paper. The introduction's stated contributions—including benchmark design, evaluation across 30 models, and empirical insights—are all directly addressed and supported by the experimental and analytical sections.

   Guidelines:

   - The answer NA means that the abstract and introduction do not include the claims made in the paper.
   - The abstract and/or introduction should clearly state the claims made, including the contributions made in the paper and important assumptions and limitations. A No or NA answer to this question will not be perceived well by the reviewers.
   - The claims made should match theoretical and experimental results, and reflect how much the results can be expected to generalize to other settings.
   - It is fine to include aspirational goals as motivation as long as it is clear that these goals are not attained by the paper.

2. **Limitations**

   Question: Does the paper discuss the limitations of the work performed by the authors?

   Answer: [Yes]

   Justification: The limitation is discussed in section 7. These reflections demonstrate awareness of the benchmark's scope and encourage future improvement.

   Guidelines:

   - The answer NA means that the paper has no limitation while the answer No means that the paper has limitations, but those are not discussed in the paper.
   - The authors are encouraged to create a separate "Limitations" section in their paper.
   - The paper should point out any strong assumptions and how robust the results are to violations of these assumptions (e.g., independence assumptions, noiseless settings, model well-specification, asymptotic approximations only holding locally). The authors should reflect on how these assumptions might be violated in practice and what the implications would be.
   - The authors should reflect on the scope of the claims made, e.g., if the approach was only tested on a few datasets or with a few runs. In general, empirical results often depend on implicit assumptions, which should be articulated.
   - The authors should reflect on the factors that influence the performance of the approach. For example, a facial recognition algorithm may perform poorly when image resolution is low or images are taken in low lighting. Or a speech-to-text system might not be used reliably to provide closed captions for online lectures because it fails to handle technical jargon.
   - The authors should discuss the computational efficiency of the proposed algorithms and how they scale with dataset size.
   - If applicable, the authors should discuss possible limitations of their approach to address problems of privacy and fairness.
   - While the authors might fear that complete honesty about limitations might be used by reviewers as grounds for rejection, a worse outcome might be that reviewers discover limitations that aren't acknowledged in the paper. The authors should use their best

judgment and recognize that individual actions in favor of transparency play an important role in developing norms that preserve the integrity of the community. Reviewers will be specifically instructed to not penalize honesty concerning limitations.

3. **Theory assumptions and proofs**

   Question: For each theoretical result, does the paper provide the full set of assumptions and a complete (and correct) proof?

   Answer: [NA]

   Justification: The paper does not present any theoretical results requiring formal assumptions or mathematical proofs. The only formulaic reference is the Bonferroni correction for multiple hypothesis testing, which is a standard statistical adjustment rather than a novel theoretical contribution.

   Guidelines:

   - The answer NA means that the paper does not include theoretical results.
   - All the theorems, formulas, and proofs in the paper should be numbered and cross-referenced.
   - All assumptions should be clearly stated or referenced in the statement of any theorems.
   - The proofs can either appear in the main paper or the supplemental material, but if they appear in the supplemental material, the authors are encouraged to provide a short proof sketch to provide intuition.
   - Inversely, any informal proof provided in the core of the paper should be complemented by formal proofs provided in appendix or supplemental material.
   - Theorems and Lemmas that the proof relies upon should be properly referenced.

4. **Experimental result reproducibility**

   Question: Does the paper fully disclose all the information needed to reproduce the main experimental results of the paper to the extent that it affects the main claims and/or conclusions of the paper (regardless of whether the code and data are provided or not)?

   Answer: [Yes]

   Justification: The paper provides a comprehensive and transparent account of all components necessary to reproduce its main experimental results. The MMAR dataset, including both JSON annotations and audio files, is publicly hosted on Hugging Face, and the evaluation script (evaluate.py) is available on GitHub. Detailed descriptions are provided for all 30 evaluated models that are either open-sourced or have APIS available, including citations and categorisation across five architecture types. The paper also clearly explains the evaluation procedure, including how audio captions are generated and used as inputs to non-audio models. This level of detail ensures reproducibility of both the experimental setup and the main benchmarking results.

   Guidelines:

   - The answer NA means that the paper does not include experiments.
   - If the paper includes experiments, a No answer to this question will not be perceived well by the reviewers: Making the paper reproducible is important, regardless of whether the code and data are provided or not.
   - If the contribution is a dataset and/or model, the authors should describe the steps taken to make their results reproducible or verifiable.
   - Depending on the contribution, reproducibility can be accomplished in various ways. For example, if the contribution is a novel architecture, describing the architecture fully might suffice, or if the contribution is a specific model and empirical evaluation, it may be necessary to either make it possible for others to replicate the model with the same dataset, or provide access to the model. In general. releasing code and data is often one good way to accomplish this, but reproducibility can also be provided via detailed instructions for how to replicate the results, access to a hosted model (e.g., in the case of a large language model), releasing of a model checkpoint, or other means that are appropriate to the research performed.

- While NeurIPS does not require releasing code, the conference does require all submissions to provide some reasonable avenue for reproducibility, which may depend on the nature of the contribution. For example
  (a) If the contribution is primarily a new algorithm, the paper should make it clear how to reproduce that algorithm.
  (b) If the contribution is primarily a new model architecture, the paper should describe the architecture clearly and fully.
  (c) If the contribution is a new model (e.g., a large language model), then there should either be a way to access this model for reproducing the results or a way to reproduce the model (e.g., with an open-source dataset or instructions for how to construct the dataset).
  (d) We recognize that reproducibility may be tricky in some cases, in which case authors are welcome to describe the particular way they provide for reproducibility. In the case of closed-source models, it may be that access to the model is limited in some way (e.g., to registered users), but it should be possible for other researchers to have some path to reproducing or verifying the results.

5. **Open access to data and code**

   Question: Does the paper provide open access to the data and code, with sufficient instructions to faithfully reproduce the main experimental results, as described in supplemental material?

   Answer: [Yes]

   Justification: The paper provides open access to both the dataset and evaluation code. The MMAR benchmark, including audio files and JSON annotations, is released on Hugging Face, while the evaluation script (evaluate.py) and usage instructions are hosted on GitHub.

   Guidelines:

   - The answer NA means that paper does not include experiments requiring code.
   - Please see the NeurIPS code and data submission guidelines (`https://nips.cc/public/guides/CodeSubmissionPolicy`) for more details.
   - While we encourage the release of code and data, we understand that this might not be possible, so "No" is an acceptable answer. Papers cannot be rejected simply for not including code, unless this is central to the contribution (e.g., for a new open-source benchmark).
   - The instructions should contain the exact command and environment needed to run to reproduce the results. See the NeurIPS code and data submission guidelines (`https://nips.cc/public/guides/CodeSubmissionPolicy`) for more details.
   - The authors should provide instructions on data access and preparation, including how to access the raw data, preprocessed data, intermediate data, and generated data, etc.
   - The authors should provide scripts to reproduce all experimental results for the new proposed method and baselines. If only a subset of experiments are reproducible, they should state which ones are omitted from the script and why.
   - At submission time, to preserve anonymity, the authors should release anonymized versions (if applicable).
   - Providing as much information as possible in supplemental material (appended to the paper) is recommended, but including URLs to data and code is permitted.

6. **Experimental setting/details**

   Question: Does the paper specify all the training and test details (e.g., data splits, hyper-parameters, how they were chosen, type of optimizer, etc.) necessary to understand the results?

   Answer: [Yes]

   Justification: The paper specifies all relevant experimental details necessary to understand and reproduce the results. The dataset curation process is extensively documented, including how questions, answers, and chains-of-thought were created, validated, and quality-checked. The evaluation setting clearly defines the task formulation (multiple-choice), metric (classification accuracy), and model input structure (audio + question + choices). For cascaded

models, the paper describes how audio is first converted to captions before being fed into language models. Although the paper does not involve model training and therefore omits hyperparameter or optimizer details, it thoroughly covers all required test-time settings. Model categories, individual model sources, and evaluation procedures are well-documented.

Guidelines:

- The answer NA means that the paper does not include experiments.
- The experimental setting should be presented in the core of the paper to a level of detail that is necessary to appreciate the results and make sense of them.
- The full details can be provided either with the code, in appendix, or as supplemental material.

7. **Experiment statistical significance**

Question: Does the paper report error bars suitably and correctly defined or other appropriate information about the statistical significance of the experiments?

Answer: [Yes]

Justification: The paper reports statistical significance using a Bonferroni correction to account for multiple hypothesis testing across 30 models. Specifically, it sets a per-model significance threshold of $p < 0.001$, resulting in a family-wise error rate of 0.03. This approach is appropriate for controlling the false positive rate under multiple comparisons and aligns with standard practices in statistical evaluation. While the paper does not include confidence intervals or error bars for each model's accuracy, the application of corrected p-values provides sufficient information to assess the reliability of the reported performance differences across models.

Guidelines:

- The answer NA means that the paper does not include experiments.
- The authors should answer "Yes" if the results are accompanied by error bars, confidence intervals, or statistical significance tests, at least for the experiments that support the main claims of the paper.
- The factors of variability that the error bars are capturing should be clearly stated (for example, train/test split, initialization, random drawing of some parameter, or overall run with given experimental conditions).
- The method for calculating the error bars should be explained (closed form formula, call to a library function, bootstrap, etc.)
- The assumptions made should be given (e.g., Normally distributed errors).
- It should be clear whether the error bar is the standard deviation or the standard error of the mean.
- It is OK to report 1-sigma error bars, but one should state it. The authors should preferably report a 2-sigma error bar than state that they have a 96% CI, if the hypothesis of Normality of errors is not verified.
- For asymmetric distributions, the authors should be careful not to show in tables or figures symmetric error bars that would yield results that are out of range (e.g. negative error rates).
- If error bars are reported in tables or plots, The authors should explain in the text how they were calculated and reference the corresponding figures or tables in the text.

8. **Experiments compute resources**

Question: For each experiment, does the paper provide sufficient information on the computer resources (type of compute workers, memory, time of execution) needed to reproduce the experiments?

Answer: [Yes]

Justification: Information discussed in Appendix L. While exact memory and storage configurations are not specified for each run, the provided information is sufficient to estimate the compute scale and replicate the experimental setup.

Guidelines:

- The answer NA means that the paper does not include experiments.

- The paper should indicate the type of compute workers CPU or GPU, internal cluster, or cloud provider, including relevant memory and storage.
- The paper should provide the amount of compute required for each of the individual experimental runs as well as estimate the total compute.
- The paper should disclose whether the full research project required more compute than the experiments reported in the paper (e.g., preliminary or failed experiments that didn't make it into the paper).

9. **Code of ethics**

Question: Does the research conducted in the paper conform, in every respect, with the NeurIPS Code of Ethics `https://neurips.cc/public/EthicsGuidelines`?

Answer: [Yes]

Justification: The research conducted in the paper conforms to the NeurIPS Code of Ethics in all respects, including ethical data sourcing, fair labor practices, legal compliance, privacy protection, and responsible dissemination in Appendix K By adhering to these ethical standards throughout data collection, annotation, and release, the work aligns with the spirit and letter of the NeurIPS Code of Ethics.

Guidelines:

- The answer NA means that the authors have not reviewed the NeurIPS Code of Ethics.
- If the authors answer No, they should explain the special circumstances that require a deviation from the Code of Ethics.
- The authors should make sure to preserve anonymity (e.g., if there is a special consideration due to laws or regulations in their jurisdiction).

10. **Broader impacts**

Question: Does the paper discuss both potential positive societal impacts and negative societal impacts of the work performed?

Answer: [NA]

Justification: The paper presents a benchmark dataset for academic research and does not involve the development or deployment of models with direct real-world applications. It does not include personal data, sensitive content, or generative components, and all audio clips are sourced from publicly available material and limited to under 30 seconds. Given its foundational nature and restricted non-commercial license (CC-BY-NC), the work does not pose immediate societal impact risks.

Guidelines:

- The answer NA means that there is no societal impact of the work performed.
- If the authors answer NA or No, they should explain why their work has no societal impact or why the paper does not address societal impact.
- Examples of negative societal impacts include potential malicious or unintended uses (e.g., disinformation, generating fake profiles, surveillance), fairness considerations (e.g., deployment of technologies that could make decisions that unfairly impact specific groups), privacy considerations, and security considerations.
- The conference expects that many papers will be foundational research and not tied to particular applications, let alone deployments. However, if there is a direct path to any negative applications, the authors should point it out. For example, it is legitimate to point out that an improvement in the quality of generative models could be used to generate deepfakes for disinformation. On the other hand, it is not needed to point out that a generic algorithm for optimizing neural networks could enable people to train models that generate Deepfakes faster.
- The authors should consider possible harms that could arise when the technology is being used as intended and functioning correctly, harms that could arise when the technology is being used as intended but gives incorrect results, and harms following from (intentional or unintentional) misuse of the technology.
- If there are negative societal impacts, the authors could also discuss possible mitigation strategies (e.g., gated release of models, providing defenses in addition to attacks, mechanisms for monitoring misuse, mechanisms to monitor how a system learns from feedback over time, improving the efficiency and accessibility of ML).

11. **Safeguards**

    Question: Does the paper describe safeguards that have been put in place for responsible release of data or models that have a high risk for misuse (e.g., pretrained language models, image generators, or scraped datasets)?

    Answer: [NA]

    Justification: The paper does not release any models or data with high risk for misuse. The MMAR benchmark contains short (<30s) audio clips sourced from publicly available videos, with no personal or sensitive content. It does not involve the release of generative models or systems that could be repurposed for harmful applications. As such, safeguards for high-risk assets are not necessary in this context.

    Guidelines:

    - The answer NA means that the paper poses no such risks.
    - Released models that have a high risk for misuse or dual-use should be released with necessary safeguards to allow for controlled use of the model, for example by requiring that users adhere to usage guidelines or restrictions to access the model or implementing safety filters.
    - Datasets that have been scraped from the Internet could pose safety risks. The authors should describe how they avoided releasing unsafe images.
    - We recognize that providing effective safeguards is challenging, and many papers do not require this, but we encourage authors to take this into account and make a best faith effort.

12. **Licenses for existing assets**

    Question: Are the creators or original owners of assets (e.g., code, data, models), used in the paper, properly credited and are the license and terms of use explicitly mentioned and properly respected?

    Answer: [Yes]

    Justification: All existing assets used in the paper, including pre-trained models and datasets, are properly cited with references to their original publications. The terms of use for each asset are respected, and our benchmark is released under the MIT license. Audio data is sourced from publicly available, user-uploaded internet content, and is used under fair-use considerations with additional restrictions (e.g., <30 seconds, CC-BY-NC license) to minimize copyright risks.

    Guidelines:

    - The answer NA means that the paper does not use existing assets.
    - The authors should cite the original paper that produced the code package or dataset.
    - The authors should state which version of the asset is used and, if possible, include a URL.
    - The name of the license (e.g., CC-BY 4.0) should be included for each asset.
    - For scraped data from a particular source (e.g., website), the copyright and terms of service of that source should be provided.
    - If assets are released, the license, copyright information, and terms of use in the package should be provided. For popular datasets, `paperswithcode.com/datasets` has curated licenses for some datasets. Their licensing guide can help determine the license of a dataset.
    - For existing datasets that are re-packaged, both the original license and the license of the derived asset (if it has changed) should be provided.
    - If this information is not available online, the authors are encouraged to reach out to the asset's creators.

13. **New assets**

    Question: Are new assets introduced in the paper well documented and is the documentation provided alongside the assets?

    Answer: [Yes]

Justification: The paper introduces a new benchmark (MMAR), and all associated assets—including audio files, annotations, and evaluation scripts—are well documented in the README files of both the Hugging Face dataset repository and the GitHub codebase.

Guidelines:

- The answer NA means that the paper does not release new assets.
- Researchers should communicate the details of the dataset/code/model as part of their submissions via structured templates. This includes details about training, license, limitations, etc.
- The paper should discuss whether and how consent was obtained from people whose asset is used.
- At submission time, remember to anonymize your assets (if applicable). You can either create an anonymized URL or include an anonymized zip file.

14. **Crowdsourcing and research with human subjects**

    Question: For crowdsourcing experiments and research with human subjects, does the paper include the full text of instructions given to participants and screenshots, if applicable, as well as details about compensation (if any)?

    Answer: [Yes]

    Justification: The paper does not involve crowdsourcing or external human subject experiments. All annotations were performed by the authors themselves. However, to ensure transparency, the paper includes screenshots of the annotation interface in Appendix C and Appendix D and provides documentation of the labeling process.

    Guidelines:

    - The answer NA means that the paper does not involve crowdsourcing nor research with human subjects.
    - Including this information in the supplemental material is fine, but if the main contribution of the paper involves human subjects, then as much detail as possible should be included in the main paper.
    - According to the NeurIPS Code of Ethics, workers involved in data collection, curation, or other labor should be paid at least the minimum wage in the country of the data collector.

15. **Institutional review board (IRB) approvals or equivalent for research with human subjects**

    Question: Does the paper describe potential risks incurred by study participants, whether such risks were disclosed to the subjects, and whether Institutional Review Board (IRB) approvals (or an equivalent approval/review based on the requirements of your country or institution) were obtained?

    Answer: [NA]

    Justification: All data collection and annotation were conducted by the authors and qualified collaborators. According to institutional guidelines, the study does not involve human subjects in a manner that requires IRB or ethics board approval. No direct interaction with participants occurred, and no personally identifiable information was collected.

    Guidelines:

    - The answer NA means that the paper does not involve crowdsourcing nor research with human subjects.
    - Depending on the country in which research is conducted, IRB approval (or equivalent) may be required for any human subjects research. If you obtained IRB approval, you should clearly state this in the paper.
    - We recognize that the procedures for this may vary significantly between institutions and locations, and we expect authors to adhere to the NeurIPS Code of Ethics and the guidelines for their institution.
    - For initial submissions, do not include any information that would break anonymity (if applicable), such as the institution conducting the review.

16. **Declaration of LLM usage**

Question: Does the paper describe the usage of LLMs if it is an important, original, or non-standard component of the core methods in this research? Note that if the LLM is used only for writing, editing, or formatting purposes and does not impact the core methodology, scientific rigorousness, or originality of the research, declaration is not required.

Answer: [Yes]

Justification: LLMs were used to generate distractor options for multiple-choice questions, beyond basic writing or formatting. All generated content was independently reviewed by two annotators to ensure quality. This non-standard use is clearly documented and aligns with NeurIPS guidelines.

Guidelines:

- The answer NA means that the core method development in this research does not involve LLMs as any important, original, or non-standard components.
- Please refer to our LLM policy (`https://neurips.cc/Conferences/2025/LLM`) for what should or should not be described.

# A   Task Layer Definitions

To support fine-grained evaluation of reasoning depth, MMAR organizes task categories into a four-layer hierarchical taxonomy, progressing from low-level signal analysis to high-level cultural understanding. Below, we provide formal definitions for each layer, as well as examples from their sub-categories:

**Signal Layer.**    This layer focuses on the direct analysis of raw acoustic features such as frequency, amplitude, duration, rhythm, or silence. Tasks at this level require minimal semantic context and are grounded in low-level signal recognition.

> **Examples**
>
> Detecting whether a sound is continuous or intermittent; identifying pitch change; recognizing audio distortions or reverberations.

| Sub-Category | Metadata |
|---|---|
| **Acoustic Quality Analysis** | **Question:**
The part of the metal ruler extending from the table is moved, during which attempt is the extended length the longest?
**Choices & Answer:**
A. First time.
B. Second to last time.
C. Last time.
D. Second time
**CoT:**
The longer the extended length of the steel ruler, the lower its vibration frequency and pitch. Thus, the final extension, which produces the lowest pitch, is the longest. |
| **Anomaly Detection** | **Question:**
Is the scream in the audio from the music?
**Choices & Answer:**
A. Yes.
B. No.
**CoT:**
The scream is clearly separate from the music—it sounds like someone shouting and is noticeably out of harmony. |
| **Audio Difference Analysis** | **Question:**
What type of keyboard made the first sound?
**Choices & Answer:**
A. Tactile.
B. Silent.
C. Linear.
D. Clicky.
**CoT:**
The first keyboard produces a loud click sound with a noticeable bump during the keypress, typical of clicky switches. |

**Perception Layer.**    This layer requires the model to interpret perceptual patterns in audio, such as identifying the type of sound, the speaker's paralinguistic features, or the instrumentation in music. These tasks depend on human-like perceptual abstraction beyond raw signal analysis.

> **Examples**
>
> Recognizing whether a voice sounds like; identifying the presence of a crowd; distinguishing between classical and electronic music.

| Sub-Category | Metadata |
|---|---|
| **Spatial Analysis** | **Question:**
Is the boat approaching or moving away?
**Choices & Answer:**
A. Approaching.
B. Moving away.
**CoT:**
It begins with a loud horn, followed by crashing and metallic squeezing sounds, all at high volume. These are immediately followed by screams from a nearby crowd, indicating the incident occurred close to the listener. This suggests the boat is moving toward the recorder or the crowd. |
| **Temporal Analysis** | **Question:**
Where is the sports game being watched?
**Choices & Answer:**
A. On the radio.
B. On a mobile phone.
C. At the stadium.
D. On a television.
**CoT:**
The sounds of the game appear in both segments, with a remote control click clearly heard in between. This suggests the game is being watched on a television rather than at a live venue. |
| **Correlation Analysis** | **Question:**
Does the person in the audio find the hotpot spicy?
**Choices & Answer:**
A. Not very spicy.
B. Very spicy.
**CoT:**
The speaker mentions they are attempting the spiciest hotpot, and after eating, they make panting and coughing sounds — clear signs of being overwhelmed by the spiciness. |
| **Counting and Statistics** | **Question:**
This is the sound of a potato being chopped. Into how many pieces is the potato cut?
**Choices & Answer:**
A. 15.
B. 8.
C. 9.
D. 12.
**CoT:**
The consistent sound of a knife hitting a cutting board suggests sequential slicing. Eight distinct chopping sounds are heard, which means the potato was cut 8 times, resulting in 9 pieces. |
| **Music Theory** | **Question:**
Identify the musical period.
**Choices & Answer:**
A. Romantic period.
B. Classical period.
C. Baroque period.
D. Modern period.
**CoT:**
The lyrical and expressive melody, rich harmonic progressions, and tonal modulations are characteristic of the Romantic period. The piece also resembles the style of Chopin, further supporting this classification. |

| Sub-Category | Metadata |
|---|---|
| **Environmental Perception and Reasoning** | **Question:**
Where did this happen?
**Choices & Answer:**
A. Airport.
B. Supermarket.
C. Hotel.
D. Bank.
**CoT:**
The audio begins with a thud followed by a man urgently instructing someone to raise their hands and place 'clean unmarked bills' into a bag—typical cues associated with a robbery. Additional details like 'tap that pile of receipts,' 'forget about the money,' and 'fix that notepad so it's in the right angle with the corner of your desk' point to a transactional setting with a counter and office supplies, all consistent with a bank environment. |

**Semantic Layer.** Tasks in this layer require understanding the meaning or intent behind audio content. The model must perform reasoning based on linguistic content (in speech), sound semantics (in events), or cross-modal grounding.

> **Examples**
>
> Inferring the action described in speech; understanding whether an audio clip describes a danger; answering what event is taking place based on complex audio scenes.

| Sub-Category | Metadata |
|---|---|
| **Content Analysis** | **Question:**
What is Gray's mother's name?
**Choices & Answer:**
A. Emily.
B. Lisa.
C. Mary.
D. Unknown.
**CoT:**
The speaker (possibly a police officer) is helping a child named Gray find his mom. When asked, Gray only refers to her as 'Mommy,' providing no name. Thus, her name remains unknown. |
| **Emotion and Intention** | **Question:**
Who is missing?
**Choices & Answer:**
A. Little Panda.
B. Tiny Elephant.
C. Big Koala.
D. Little Koala.
The speaker calls out 'Little Koala' twice, followed by a startled non-verbal reaction, suggesting that Little Koala is absent during a roll call or check. |
| **Speaker Analysis** | **Question:**
Who is faster now?
**Choices & Answer:**
A. Ray.
B. Tayo.
C. Shine.
D. Speedy.
**CoT:**
Shine mentioned that he installed new tires and can go faster now. His name was mentioned by Tayo during the greeting at the beginning. |

**Cultural Layer.** This layer evaluates higher-order reasoning grounded in social, cultural, or contextual knowledge. It requires the integration of audio with world knowledge, social norms, or domain-specific expertise.

> **Examples**
>
> Understanding the symbolic meaning of a national anthem; distinguishing humor vs sarcasm in tone; recognizing culturally specific sounds like instruments, rituals, or festivals.

| Sub-Category | Metadata |
|---|---|
| **Culture of Speaker** | **Question:**
How many different Chinese tones are demonstrated across the six syllables?
**Choices & Answer:**
A. 3.
B. 2.
C. 5.
D. 4.
**CoT:**
The speaker pronounces: ma (first tone), ma (first tone), qi (second tone), ma (third tone), ma (third tone), and man (fourth tone). This covers all four Mandarin tones. |
| **Imagination** | **Question:**
What would he have seen if he had arrived earlier?
**Choices & Answer:**
A. Piano performance.
B. Harp performance.
C. Drum performance.
D. Gong strike.
**CoT:**
The speaker runs and bumps into a closed door, sighs, and says 'Open the door, let me in.' Drum sounds are then heard from inside, suggesting he just missed a drumming event. |
| **Aesthetic Analysis** | **Question:**
Among the four piano passages in the audio, which one is the best?
**Choices & Answer:**
A. The fourth passage.
B. The second passage.
C. The first passage.
D. The third passage.
**CoT:**
The first passage is a simple single-note melody with a steady rhythm but lacks harmonic depth. The second adds chordal accompaniment, though the left hand is highly repetitive and technically basic. The third introduces rhythmic complexity and shows moderate technical difficulty. The fourth combines complex rhythms, arpeggiated chords, and fast note clusters, clearly demonstrating the performer's control and speed—making it the most musically and technically impressive. |

# B Annotator Team Qualifications

## B.1 Speech and Sound Tasks

For speech and sound tasks, all annotation, correction, and quality assurance personnel possessed strong academic and research backgrounds. Every team member held at least a bachelor's degree, with over half either currently enrolled in or having completed a Ph.D. program. All annotators had a minimum of one year of research experience in relevant areas such as speech processing, acoustic

modeling, or audio understanding. In addition, the majority had publication experience in top-tier conferences (e.g., ICASSP, INTERSPEECH, NeurIPS, ICML), ensuring a deep understanding of both the signal and semantic aspects of audio content.

## B.2 Music Tasks

For music-related tasks, the team included a mix of experts from both academic and artistic backgrounds. One group consisted of researchers in music technology, all of whom had published in prominent venues such as ISMIR. The second group included graduate students and alumni from prestigious music conservatories, including the Central Conservatory of Music (China) and Carnegie Mellon University School of Music, bringing expert-level music perception and domain knowledge. The annotation team also included professionals with industry experience, such as trained annotators from commercial annotation companies and former annotation staff from AI music companies, further contributing practical expertise and quality assurance.

## C Data Annotation Platform

We conducted annotation correction and quality inspection using a professional annotation platform that supports structured editing, version control, and multi-stage review. All fields of each question, including question, choice, answer, reasoning chain, and other metadata were reviewed and corrected through this system. A Snapshot of the platform interface is shown in Figure 7.

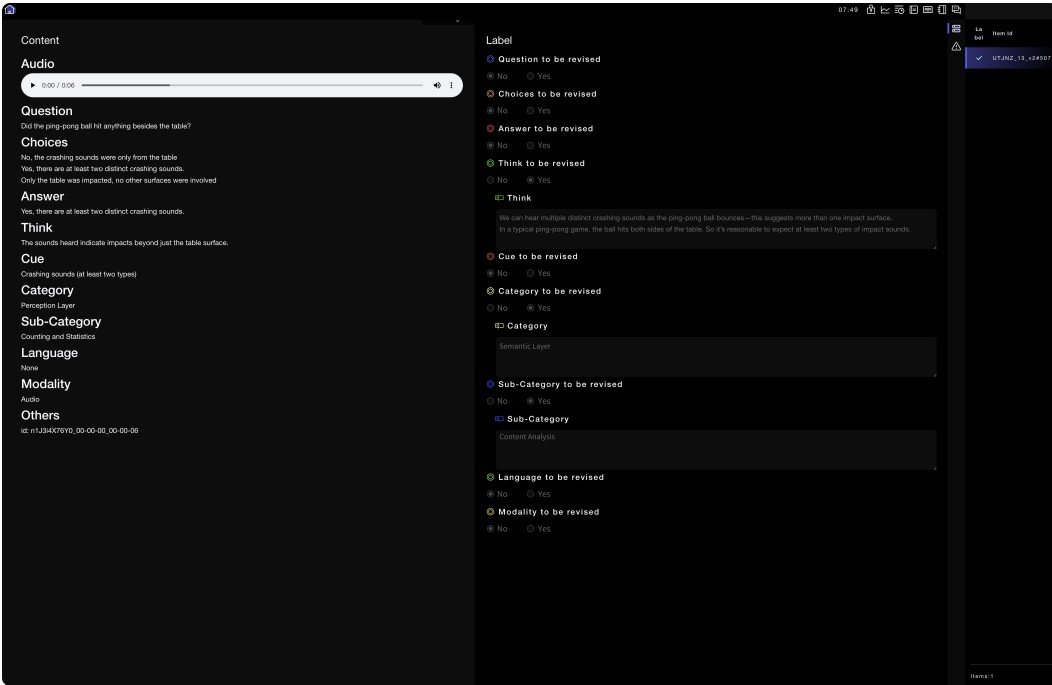

Figure 7: A Snapshot of the data annotation platform for correction and quality inspection.

## D Data Annotation Instruction Document

To ensure the quality and depth of MMAR, we provided annotators with the following instructions:

1. **Source and Duration Requirements**
   (a) All audio must be sourced from **real-world internet videos**, with **video links and timestamps** preserved.
   (b) For other sources (e.g., symbolic music rendering), confirm with the dataset coordinator.
   (c) Each audio clip must be no longer than **30 seconds**.

2. **Language and Expression**

   (a) Q&A can be written in Chinese during drafting but should be finalized in English where appropriate.

   (b) Use English for external references or precise expressions.

   (c) Avoid specific names unless the person's identity is crucial and deducible from audio.

   (d) Prefer role-based descriptors (e.g., "scorer", "driver") over personal names.

3. **Modal Relevance**

   (a) All questions must depend on **audio modality alone** and not be answerable from language priors.

   (b) Annotators should close their eyes and verify that answers can only be inferred from the audio.

   (c) Avoid intuitive or guessable questions (e.g., "What animal barks?" → "Dog").

4. **Answer Quality**

   (a) Answers should be **concise**, ideally just a few words or phrases.

   (b) They must be **objective or subjectively universal**.

   (c) Ensure that each question has only one valid answer and no alternative interpretations.

   (d) Avoid overly broad answers or unclear formulations.

5. **Reasoning Requirements**

   (a) Each question must involve at least **two reasoning cues**, including one from audio.

   (b) Require **multi-step reasoning**; questions solvable by direct captioning or ASR should be avoided.

   (c) Avoid:
   - Questions based on a single common sound or simple object recognition.
   - Use of specific real-world names, figures, landmarks, or artifacts.
   - Instead, use descriptive terms (e.g., "a tall clock tower with a chime" instead of "Big Ben").

6. **Creativity and Cognitive Challenge**

   (a) Encourage questions that challenge **knowledge, logic, perception, and reasoning**.

   (b) Include higher-order formats such as **retrocausal, predictive, counterfactual, evaluative**, or **planning** questions.

   (c) Introduce domain-specific knowledge, such as physics, music, sports, or social reasoning.

# E  Benchmarking Candidates

## E.1  Large Audio Language Models (LALMs)

**Audio Flamingo.**   Developed by NVIDIA, Audio Flamingo augments a large language model with an audio frontend, enabling open-ended audio understanding of non-speech sounds. It integrates a pretrained audio encoder with a text decoder LLM and supports retrieval-based few-shot prompting and multi-turn dialogue. The model demonstrated state-of-the-art performance across diverse audio QA and captioning tasks and is open-source.

**Audio Flamingo 2.**   A collaboration between NVIDIA and UMD, AF2 enhances long-audio reasoning (up to 5 minutes) via a 3B LLM, custom CLAP encoder, and a multi-stage training curriculum. It introduces two reasoning-focused datasets and outperforms larger proprietary models on complex audio tasks. The model and checkpoints are open-source.

**LTU.**   From MIT-IBM, LTU integrates a frozen Vicuna model with an audio encoder using LoRA adapters, trained on OpenAQA for non-speech audio QA. It processes continuous audio tokens for open-ended reasoning and is open-source with weights and code.

**LTU-AS.** An extension of LTU for both speech and non-speech audio. It incorporates Whisper for speech recognition and was trained on OpenASQA. LTU-AS jointly models speech transcription and audio scene understanding, offering robust multimodal audio perception in one system. Open-source.

**MusiLingo.** Developed by a multi-institutional team, MusiLingo aligns music audio with language using MERT and Vicuna. It excels in music captioning and Q&A, trained on MusicCaps and MusicInstruct datasets. Open-source with code and models.

**MuLLaMA.** A music-focused model combining understanding and generation. MuLLaMA extends LLaMA with music audio, symbolic data, and image/video context, supporting tasks from genre recognition to music generation. Fully open-source.

**GAMA.** From UMD and Adobe, GAMA uses an Audio Q-Former module to produce audio tokens from AST features. It achieves strong results on reasoning-intensive audio benchmarks via instruction-tuning on a synthetic reasoning dataset. Open-source with demo.

**Qwen-Audio.** Alibaba's LALM trained on over 30 audio tasks with a hierarchical tagging scheme. It supports speech, music, and environmental audio in a multi-task framework. Open-source with chat capabilities.

**Qwen2-Audio.** Successor to Qwen-Audio with simplified prompting and two interaction modes: Voice Chat and Audio Analysis. It automatically switches modes and excels on the AIR-Bench benchmark. Open-source.

**SALMONN.** A unified audio-language model supporting speech, environmental sound, and music. It uses dual encoders and covers diverse tasks. Demonstrates emergent capabilities via activation tuning. Fully open-source.

**GPT-4o Mini Audio.** A smaller variant of GPT-4o offering real-time speech input/output and asynchronous audio tasks. Despite reduced size, it retains strong transcription and generation ability. API-accessible.

**GPT-4o Audio.** OpenAI's flagship multimodal model supporting audio, vision, and text. It processes audio in real time and produces spoken responses with high fidelity, surpassing prior SOTA in ASR and dialogue. Proprietary but widely deployed.

## E.2 Large Audio Reasoning Models (LARMs)

**Mellow.** A compact 167M-parameter audio-language model optimized for reasoning tasks. Trained on 155 hours of audio and the ReasonAQA dataset, it matches or surpasses larger models like Qwen2-Audio on benchmarks such as MMAU, demonstrating the potential of small ALMs for efficient, audio-grounded reasoning. Open-source.

**Audio-CoT.** A methodology applying chain-of-thought reasoning to LALMs. Improves performance on reasoning tasks via guided intermediate steps. Offers insight into prompt design for better auditory reasoning. Not a model but a strategy.

**Audio-Reasoner.** A reasoning-specialized model fine-tuned on a CoT-annotated dataset (CoTA) with structured multi-step reasoning. Achieves state-of-the-art on multiple benchmarks, demonstrating the value of reasoning-rich supervision. Open-source.

## E.3 Omni Language Models (OLMs)

**AnyGPT.** A unified multimodal LLM using discrete tokenization to represent and process audio, image, speech, and text. Trained on AnyInstruct dialogues, supports any-to-any generation. Fully open-source.

**OpenOmni.** An open omnimodal LLM supporting speech, vision, and text. Uses progressive alignment and speech generation modules for real-time multimodal interaction. Achieves SOTA among open models. Open-source with demos.

**Baichuan-Omni-1.5B.** A lightweight omnimodal model optimized for edge deployment. Supports all four modalities with strong performance in a compact form. Enables real-time audio and video processing on mobile platforms. Open-source.

**Qwen 2.5 Omni.** Alibaba's dual-path "Thinker–Talker" model supporting real-time multimodal interaction. Handles text, image, audio, and video with streaming capabilities. Excels in dialogue, translation, and reasoning. Open-source.

**Gemini 2.0 Flash.** Google's advanced multimodal model with ultra-low latency and massive context support. Handles complex audio-visual tasks in real-time. Proprietary, pushing the frontier of interactive AI.

### E.4 Large Language Models (LLMs)

**DeepSeek-V3.** A 671B MoE model with multimodal support and exceptional math/coding reasoning. Accepts audio/image transcripts and performs high-accuracy inference. Fully open-source.

**GPT-4o.** OpenAI's top general-purpose model supporting text, image, and audio. It can reason over audio descriptions and engage in complex multimodal dialogue. Proprietary and deployed in multiple interfaces.

### E.5 Large Reasoning Models (LRMs)

**DeepSeek-R1.** A reasoning-optimized LLM trained with reinforcement learning and chain-of-thought strategies. Excels at multi-step logic, including audio transcript reasoning. Open-source with restrictions.

**DeepSeek-O1.** An early experimental model from DeepSeek for basic multimodal input via captions. Enables simple audio/image integration for reasoning tasks.

**DeepSeek-O3.** A later-stage open omni-modal model combining MoE structure with discrete token-based multimodal processing. Expected to handle vision, audio, and text at scale with strong reasoning capabilities.

## F Comparison with MMAU

To further illustrate the difficulty of MMAR, we compare the performance of several representative models on MMAR and the widely-used MMAU benchmark (test-mini split). As shown in Figure 8, all models experience a significant drop in accuracy when evaluated on MMAR. For instance, while Qwen-2.5-Omni 7B achieves over 65% accuracy on MMAU, its performance drops below 60% on MMAR. Greater performance differences are observed for both Audio-Reasoner and Qwen2-Audio-Instruct. This gap highlights that MMAR poses a substantially more challenging set of tasks, requiring deeper reasoning, more nuanced perception, and broader knowledge across audio domains.

## G Error Analysis

**Perceptual Errors** (37%) constitute the most frequent failure type, revealing core challenges in decoding music, speech, and sound events. These include misclassification of environmental sounds (e.g., mistaking wind and ice cracking for light rowing), under-detection of repeated audio events, and confusion in musical elements (e.g., miscounting instrument hits or players, misinterpreting pitch, mode, or performance techniques). Other cases involve errors in speech perception such as inaccurate ASR transcriptions, dialect confusion (e.g., Sichuan vs. Northeastern Mandarin), or neglecting speaker tone and emotion. Notably, many failures stem from overemphasis on isolated

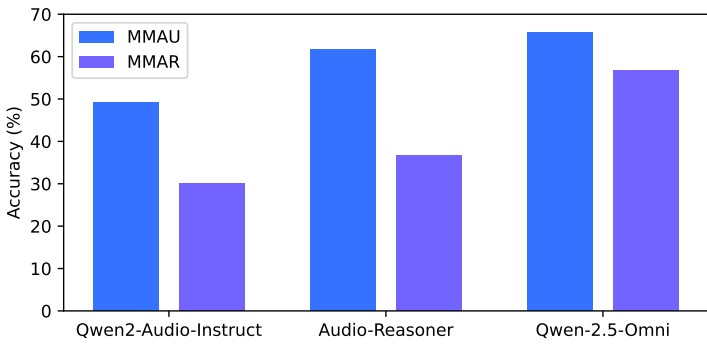

Figure 8: Different Audio-Language models' performance on MMAU (test-mini) and MMAR.

events or textual content, while overlooking global auditory context such as reverberation, speaker count, or recording quality. Additionally, the model struggles with polyphonic disambiguation and fine-grained music information retrieval. These patterns highlight the need for enhanced auditory scene analysis and temporal reasoning capabilities.

**Lack of Knowledge** (9%) Knowledge-related failures arise when the model correctly perceives the sound but lacks the necessary domain understanding to interpret it. These include errors in commonsense reasoning (e.g., assuming soup can be made with oil instead of water), cultural grounding (e.g., confusing the morin khuur with the violin, or failing to infer character relationships in Harry Potter), and scientific knowledge (e.g., linking free-fall duration to splash timing or failing to decode Morse code). Such errors reveal that the model's latent representations are insufficiently grounded in auditory semantics and domain-specific concepts, underscoring the need for curated multimodal pretraining datasets enriched with audio-centric factual knowledge.

**Reasoning Errors** (20%) Reasoning failures occur when the model perceives relevant cues but misinterprets causal structure, temporal dynamics, or discourse-level logic. These include incorrect attribution of emotions (e.g., interpreting screams as excitement rather than fear), misunderstanding sarcasm or indirect speech (e.g., taking pranks literally), and failing to resolve coreference in conversation. Common issues also arise in quantitative reasoning (e.g., estimating event counts or durations), physical plausibility (e.g., believing a balloon can be popped multiple times), and logical inference (e.g., mislocalizing screams to a train instead of a rollercoaster). These cases indicate that while the model may have access to perceptual and factual cues, it lacks consistent multi-hop reasoning capabilities across temporal and multimodal contexts. Overall, they underscore the inherent difficulty of integrating perception with high-level reasoning and point to the need for architectures capable of contextual simulation and counterfactual inference.

**Other Errors**(34%) captures diverse failure modes outside core perception, knowledge, or reasoning, and accounts for a substantial 34% of all errors. First, instruction misinterpretation (3%) reflects semantic confusion about task framing—for instance, mistaking "cutting" as an interruption in speech rather than a physical sound event, or confusing pitch class with rhythmic notation. Second, correct answers with incorrect choice selection (15%) arise when the model's rationale aligns with the correct label, but the final answer deviates due to inattentive grounding or heuristic bias (e.g., selecting an adjacent option or generating a new one not among choices). Third, generation collapse (12%) includes repetitive chain-of-thought outputs with no conclusion, often observed in abstract or ambiguous music and speech questions, revealing decoding instability in long-horizon reasoning. Finally, choice formatting errors (4%) occur when the model produces free-form responses instead of adhering to the expected A/B/C/D schema. Collectively, these errors underscore the model's fragility in instruction following and constrained decoding, suggesting the need for stronger alignment between reasoning, choice grounding, and structured output formatting.

# H   More Detailed Results

Table 5 shows combinations of different audio captioning models and LLMs on the MMAR benchmark. Table 6 shows results with noise input instead of audio input on the MMAR benchmark.

Table 5: MMAR Benchmark results with different audio captioning models and LLMs.

| Models | Size | Single Modality (%) | | | Mixed Modalities (%) | | | | Avg (%) |
|---|---|---|---|---|---|---|---|---|---|
| | | Sound | Music | Speech | Sound-Music | Sound-Speech | Music-Speech | Sound-Music-Speech | |
| Qwen2-Audio-Instruct + GPT-4o | - | 46.06 | 40.29 | 60.88 | 27.27 | 53.67 | 46.34 | 45.83 | 50.70 |
| Qwen2.5-Omni + GPT-4o | - | 43.64 | 43.69 | 64.29 | 27.27 | 51.83 | 53.66 | 29.17 | 51.80 |
| Qwen2-Audio-Instruct + OpenAI o1 | - | 48.48 | 43.20 | 63.61 | 18.18 | 56.88 | 45.12 | 45.83 | 53.00 |
| Qwen2.5-Omni + OpenAI o1 | - | 44.85 | 48.54 | 67.69 | 18.18 | 52.75 | 57.32 | 29.17 | 54.40 |

Table 6: MMAR Benchmark results with input audio replaced by same-length noise.

| Models | Size | Single Modality (%) | | | Mixed Modalities (%) | | | | Avg (%) |
|---|---|---|---|---|---|---|---|---|---|
| | | Sound | Music | Speech | Sound-Music | Sound-Speech | Music-Speech | Sound-Music-Speech | |
| Qwen2-Audio-Instruct | 8.4B | 20.16 | 24.27 | 22.79 | 36.36 | 25.23 | 20.73 | 20.83 | 23.30 |
| Audio-Reasoner | 8.4B | 35.76 | 29.61 | 29.59 | 36.36 | 31.19 | 26.83 | 29.17 | 30.80 |
| Qwen-2.5-Omni | 3B | 35.76 | 34.95 | 40.48 | 36.36 | 32.57 | 35.37 | 29.17 | 36.10 |
| Qwen-2.5-Omni | 7B | 44.85 | 32.04 | 36.39 | 27.27 | 33.94 | 41.46 | 20.83 | 36.30 |

# I   Final Meta-Data Format

Each MMAR question is stored in a structured JSON format containing the audio path, question, answer choices, correct answer, reasoning chain, modality, task category, sub-category, language, and video source information. An example of the final meta-data format is shown below in Listing 1.

Listing 1: Sample JSON annotation of MMAR.

```json
{
    "id": "UZUbPtn01kk_00-00-30_00-00-53",
    "audio_path": "./audio/UZUbPtn01kk_00-00-30_00-00-53.wav",
    "question": "Who is faster now?",
    "choices": [
        "Ray",
        "Tayo",
        "Shine",
        "Speedy"
    ],
    "answer": "Shine",
    "think":  "Shine mentioned that he installed new tires and can go
        faster now. His name was mentioned by Tayo during the greeting
        at the beginning.",
    "modality": "speech",
    "category": "Semantic Layer",
    "sub-category": "Speaker Analysis",
    "language": "en",
    "source": "youtube",
    "url": "https://www.youtube.com/watch?v=UZUbPtn01kk",
    "timestamp": "00:00:30,00:00:53"
}
```

# J   Distractor Generation

To generate high-quality distractors for multiple-choice questions, we utilize a carefully designed prompt fed to LLM (GPT-4o). The prompt instructs the model to generate 2–3 plausible but incorrect answer options. These distractors are designed to be semantically relevant yet distinct from the correct answer, increasing the difficulty and realism of the task. The prompt for the distractors generation is shown in Listing 2.

Listing 2: Prompt for distractors generation using LLM.

```
{
"template":
"You are a professional exam question generator. Your task is to
    generate plausible but incorrect distractors based on the given
    question information. Please follow these rules strictly:
```

```
 4
 5  1. If the question is a yes/no question, generate only one distractor
       (resulting in a binary choice: yes or no).
 6  2. For all other question types, generate three distractors (resulting
        in a four-option multiple-choice question).
 7
 8  Distractors should:
 9  - Be in the same language as the question (Chinese/English).
10  - Be based on clues from the question, including the fields: question,
       answer, think, and cue.
11  - Have a similar grammatical structure to the correct answer.
12  - Appear plausible but must be factually incorrect.
13  - Be returned strictly in dictionary and list format.
14
15  Avoid:
16  - Synonyms or near-synonyms of the correct answer.
17  - Logically inconsistent content.
18  - Duplicate choices.
19  - Using any information beyond the provided input.
20
21  Example input:
22  {
23    \"question\": \"What did the man do?\",
24    \"answer\": \"Threw the child into the river\",
25    \"think\": \"The man asked if the child could swim. The child said
         no. Later, there were splashing sounds. A woman then said 'Help
         him, he can't swim,' suggesting the man threw the child into the
          water to teach him to swim.\",
26    \"cue\": \"splashing sounds|I can't swim|Help him, he can't swim\"
27  }
28
29  Example output:
30  {
31    \"distractors\": [
32      \"Threw the woman into the river\",
33      \"Jumped into the river himself\",
34      \"Pulled the child out of the river\"
35    ]
36  }
37
38  Input:
39  {input_json}
40
41  Output: "
42  }
```

## K   Ethical Statement

**Human annotation and fair wages**: All annotators involved in data creation were either legally employed research assistants with music professionals or students supported by formal scholarships who are all co-authors of the paper, and were compensated in accordance with local minimum wage regulations. This aligns with NeurIPS requirements for fair compensation of human participants.

**Data privacy and consent**: All audio clips in MMAR were extracted from publicly accessible, user-uploaded videos on platforms like YouTube. Each clip is shorter than 30 seconds—shorter than preview segments on commercial platforms like Spotify—minimizing copyright and privacy risks. No personally identifiable or sensitive user information is included.

**Licensing and responsible use**: The dataset will be released under a CC-BY-NC license, explicitly limiting its use to non-commercial academic research, in accordance with NeurIPS guidelines for ethical dataset release and copyright respect.

**Diversity and representativeness**: MMAR includes a balanced range of speech and vocal music data. At least 6.7% of the samples are labelled as female speakers or singers, though over a quarter are male, and the dataset covers multiple spoken and sung languages, including English, Chinese, Japanese, Korean, French, Italian, and German. This reflects an effort toward diversity and mitigates representational bias.

**Environmental and safety considerations**: The research poses no foreseeable risks related to safety, security, discrimination, surveillance, or environmental harm. The benchmark does not involve high-compute training or deployment of risky generative models.

## L    Experiments compute resources

The Inference experiments were conducted using A800 GPUs, with each model processing the full MMAR benchmark over a period ranging from 30 minutes to 3 hours, depending on the model size and complexity. The closed-sourced models were conducted via public APIs, which introduced variability due to network latency.

## M    Bonferroni Correction

To account for multiple hypothesis testing across 30 models, we apply the Bonferroni correction to control the family-wise error rate. Specifically, we set the significance threshold for each individual test to $p = 0.001$. Under the assumption of independence, the probability that none of the 30 tests results in a false positive is approximately:

$$(1 - 0.001)^{30} \approx 0.97$$

This leads to an overall significance level of:

$$1 - (1 - 0.001)^{30} \approx 0.03$$

Therefore, using a per-test threshold of $p = 0.001$ ensures that the overall false positive rate across all 30 models remains below 3%, maintaining statistical validity in our comparative analysis.

