# OpenReview forum: "MMAR: A Challenging Benchmark for Deep Reasoning in Speech, Audio, Music, and Their Mix"
_NeurIPS.cc/2025/Datasets_and_Benchmarks_Track — NeurIPS 2025 Datasets and Benchmarks Track poster_

### Official Review · Reviewer_66d2 · 2025-06-23

**Rating:** 5
**Confidence:** 3

**Summary:**

The paper constructs a benchmark specifically designed to evaluate the deep reasoning capabilities of ALMs. This benchmark comprises 1,000 audio-question-answer triplets, encompassing sounds, music, speech, and their hybrid modalities. The tasks are categorized into four hierarchical levels: signal, perception, semantics, and culture, with each level containing subcategories. This structure comprehensively examines the models' reasoning abilities, ranging from low-level signal analysis to high-level cultural comprehension. Chain-of-Thought annotations are provided for each question to elucidate the reasoning steps, thereby supporting subsequent research on audio reasoning. Thirty models, including LALMs, LARMs, OLMs, among others, were tested. The findings reveal significant shortcomings in the deep reasoning capabilities of existing models, with open-source models performing close to random guessing. Error analysis highlights deficiencies in the models across perception, knowledge, and reasoning dimensions, emphasizing the need for innovation in both data and algorithms for audio language reasoning.

**Additional Feedback:**

++ It is recommended to increase the sample size to over 5,000 and supplement it with rare hybrid modalities (such as sound-music-speech) and long-tail tasks (such as cross-cultural audio understanding).

++ It is recommended to introduce adversarial sample training to ensure that the problems strictly rely on audio inputs; when generating distractors through LLMs, increase options that are strongly correlated with the audio signals.

++ It is recommended to incorporate visual aids (e.g., spectrograms) or cross-modal cues at the signal level to reduce the difficulty of low-level tasks. Meanwhile, design more cross-layer fusion tasks (e.g., integrating signal analysis with semantic reasoning).

**Dataset Code Accessibility:**

Yes

**Dataset Code Comments:**

The url is accessible. And the authors provide detailed description of the benchmark.

**Ethical Considerations:**

No, there are no or only very minor ethics concerns

**Final Justification:**

I have carefully read the other comments and the rebuttal. The response solved my major concers. The url of the dataset is also accessible. Therefore, I intend to give a positive opinion.

**Limitations Weaknesses:**

++ With only 1,000 samples, there is insufficient coverage of some subcategories (such as hybrid modalities), which may limit the comprehensive evaluation of a model's generalization ability.

++ Despite undergoing quality control, there are still a small number of questions that can be answered based on linguistic priors (for example, Qwen-2.5-Omni still performs slightly better than random guessing even with noisy inputs).

++ Models exhibit significantly lower performance on signal-level tasks (such as acoustic quality analysis) compared to semantic-level tasks, as these signal-level tasks involve low-level physical characteristics, which may lead to assessment biases.

**Strengths Contributions:**

++ Unlike single-modality benchmarks (such as AudioBench), MMAR encompasses seven types of audio modalities (such as mix-sound-music), with data sourced from real-world internet videos, making it more aligned with practical application scenarios.

++ The audio data is newly collected from online videos to prevent pre-training data leakage; a five-stage pipeline (including brainstorming, taxonomy construction, manual annotation, etc.) is employed to ensure high data quality, and the annotation process involves expert review and iterative refinement.

++ A deep reasoning assessment framework has been established, where each question requires multi-step reasoning, and some tasks involve specialized knowledge (such as physics and music theory), far surpassing the superficial understanding tasks of traditional benchmarks (such as MMAU).

---

> ### Author Rebuttal · Authors · 2025-07-31
>
> Dear Reviewer 66d2,
>
> We sincerely thank the reviewer for their thorough review and valuable, constructive feedback. We are greatly encouraged that the reviewer recognized the core strengths of our work, including (i) comprehensive modality coverage, including seven real-world audio modalities; (ii) High data quality with newly collected data; (iii) Novelty of our rigorous five-stage pipeline; and (iv) significantly raises the bar compared to traditional benchmarks on audio understanding and deep reasoning. Below, we address each of the reviewer's concerns in detail.
>
> ## W1: Dataset size.
> We appreciate your concern regarding the dataset size, which touches upon a crucial aspect of our benchmark's design that we also address in our Limitations (Appendix M). Our guiding principle was **"quality over quantity"**, which is especially vital for a benchmark targeting **deep audio reasoning**. To elaborate specifically:
>
> 1. **High Cost of Annotating Deep Reasoning Tasks**: As you noted in S2, our question design process is highly intricate. **As detailed in Appendix M, finding and annotating a single data point took 10-30 minutes on average per expert, excluding the multi-stage verification and refinement process**. This effort is fundamentally different from that for shallow understanding tasks that can be generated at scale.
> 2. **High Expertise Required for Annotators**: As detailed in Appendix B, our annotation and review team was composed exclusively of domain experts, including PhDs in speech processing, audio AI researchers, and specialists from leading music conservatories. This commitment to expert-driven quality was paramount.
> 3. **Comparison with Leading Benchmarks**: Our scale is well-aligned with other highly-regarded, expert-driven benchmarks. For instance, Video-MMMU [1] contains 900 QA pairs from 300 videos, LVBench [2] has 1,549 QA pairs from 500 videos, and OmniBench [3] has 1,142 samples. In the pure reasoning domain, the AIME [4] math competition comprises only 30 problems annually. This places MMAR's scale squarely within the established norms for benchmarks where **quality and difficulty are the primary metrics of value**.
>
> As we acknowledged in our paper's Limitations section (Appendix M), *"We recognize this as a constraint on coverage and plan to expand MMAR in the future, particularly in underrepresented sub-categories and modalities."* This statement reflects our commitment to the long-term growth of this resource.
> While we are committed to its future expansion, we believe the current MMAR, as presented, already constitutes a complete and substantial contribution to the community.
>
> ## W2: Data item bias.
>
> We appreciate your insightful comment regarding the observation that a small subset of our QA still can be answered based on linguistic priors. We agree that this indicates language bias exists, a known but critical issue for any multimodal benchmark, which phenomenon we have discussed in Section 6.2. Below, we detail our three-step de-biasing pipeline, which we have now integrated into our work.
>
> ### 1. Identifying the Source of Bias
>
> **We first hypothesized that the bias most likely originates from the distractors rather than the question-answer pairs themselves**. This is because our multi-stage curation process (involving expert question setter, correctors, and quality inspectors) already included rigorous checks to ensure questions required audio grounding. The automated generation of distractors, however, can inadvertently introduce linguistic patterns that powerful LLMs might exploit. Our subsequent analysis confirmed this hypothesis.
>
> ### 2. Determining Which Questions Require Mitigation
>
> We designed an automatic protocol to detect questions requiring mitigation. **The intuition is simple: if a strong text-only model can consistently answer a question correctly without the audio, the question is biased.**
>
> We operationalized this using a statistical test:
> 1. For each question, we use a powerful text-only model (gpt4.1) to answer it $n=12$ times, using only the question and multiple-choice options.
> 2. We calculate the probability of achieving the observed accuracy or higher by chance, using a binomial distribution $X \sim Binom(n,1/c)$, where $c$ is the number of choices. This yields a p-value.
> 3. **Any question with a p-value below our significance level $\alpha = 0.02$ is flagged as biased and targeted for mitigation.** This provides a robust, data-driven method to pinpoint problematic items.
>
> ### 3. Our Approach to Removing Language Bias
>
> We employed two rewriting methods to mitigate language bias caused by distractors:
>
> 1. **Method 1: Distractor-centric Generation to Break Semantic Symmetry.**:
> Refer to [5], if every negative answer choice is generated by changing a small part of the correct answer, the LLM can detect those changes to find a “centralized” description and use that cue for its prediction.
> For a given correct answer $C$, instead of generating three distractors based on $C$, we first generate a primary distractor $D_1(C)$. We then generate the remaining distractors, $D_2(D_1)$ and $D_3(D_1)$, based on the first distractor. This breaks the semantic symmetry around the correct answer, forcing the model to evaluate each option on its own merit rather than through pattern recognition.
>
> 2.  **Method 2: Adversarial Distractor Generation.**:
> Inspired by valuable suggestion from the reviewer, we introduced adversarial distractors. For a given question, we generate an option that is highly plausible based on common-sense or real-world priors but explicitly incorrect according to the audio.
> This adversarial choice is designed such that models unable to genuinely comprehend the audio content are more likely to incorrectly select it.
>
> ### 4. Experimental Validation
>
> To prove the efficacy of our de-biasing pipeline, we re-ran the "noise input" experiment on both the original and revised MMAR versions. This setting isolates the models' reliance on text priors. The results are below:
>
> |#|Exp|Method1|Method2|Qwen-2.5-Omni-7B|Qwen-2.5-Omni-3B|
> |:--:|:--:|:--:|:--:|:--:|:--:|
> |1|Random Guess (Theoretical)|✘|✘|29.3±4.8|29.3±4.8|
> |2|Original MMAR (w/ Noise)|✘|✘|36.3|35.8|
> |3|Exp1 (w/ Noise)|✔|✘|33.4|34.1|
> |4|Exp2 (w/ Noise)|✔|✔|32.6|33.9|
>
> *Table: Impact of language bias mitigation on model performance with noise input. Line 1 shows scores of the theoretical random guess, with random range with significance level of $\alpha=0.001$.*
>
> As the table shows, the performance on the original MMAR (Line 2) was slightly above the random guess baseline, confirming the presence of some language bias. Our mitigation methods (Lines 3 and 4) systematically reduce this performance. The final, de-biased MMAR (Line 4) brings the models' performance squarely within the statistical range of random guessing, which demonstrates that models are less able to exploit linguistic shortcuts.
>
> ## W3: Regarding signal-level tasks.
> We thank the reviewer for highlighting the performance disparity across task hierarchies. **We respectfully propose that this is not an "assessment bias" but rather one of the most important findings of our study, revealing a fundamental weakness in current ALMs.**
> 1. **Revealing a Critical Capability Gap**: Signal-level tasks in MMAR (e.g., estimating ruler length from pitch) require reasoning about the underlying physics of sound generation. This is a form of "audio-physical intelligence" that goes beyond simple perception. In contrast, semantic-layer tasks can often heavily leverage the pre-trained linguistic knowledge of the underlying LLM.
> 2. **Audio Perception rather than Audio Reasoning**: The stark performance gap demonstrates that current models are primarily "Language Models with Audio Perception" rather than true "Audio Reasoning Models" grounded in the first principles of the acoustic world. MMAR is uniquely designed to expose this critical distinction. Our work thus provides a clear signal to the community about where to focus future research: on developing models that can reason from the signal up, not just from the text down.
> 3. **Revision**: We will revise the discussion in Section 6.1 to more explicitly frame this performance gap as a core scientific contribution of our paper, clarifying that it highlights a critical limitation in ALMs, rather than a bias in our benchmark design.
>
> ## Summary
> Thank you again for your insightful feedback, which has significantly improved our paper. We believe these substantial revisions, particularly the rigorous, empirically-validated de-biasing of our dataset, have directly addressed your primary concerns. We respectfully hope you will consider these significant improvements in your final assessment and would be grateful if you would reconsider raising the score.
>
>
> ## Reference
>
> [1] Hu, Kairui, et al. "Video-MMMU: Evaluating knowledge acquisition from multi-discipline professional videos." arXiv preprint arXiv:2501.13826 (2025).
>
> [2] Wang, Weihan, et al. "LVBench: An extreme long video understanding benchmark." Proc. ICCV (2025).
>
> [3] Li, Yizhi, et al. "OmniBench: Towards the future of universal omni-language models." arXiv preprint arXiv:2409.15272 (2024).
>
> [4] https://huggingface.co/datasets/opencompass/AIME2025
>
> [5] Cai, Mu, et al. "Temporalbench: Benchmarking fine-grained temporal understanding for multimodal video models." arXiv preprint arXiv:2410.10818 (2024).

---

> > ### Comment · Reviewer_66d2 · 2025-08-09
> > **Thanks for the rebuttal**
> >
> > Thans for the effors to provide the rebuttal. The rebuttal indeed addressed most of my concerns. Some minior weaknesses may be solved in the final version, such as the writing, the symbols, etc. I have re-reviewed the paper.

---

### Official Review · Reviewer_R18W · 2025-07-02

**Rating:** 4
**Confidence:** 3

**Summary:**

The paper introduces MMAR, the first benchmark specifically designed to evaluate the reasoning capabilities of audio-language models (LALMs) on speech-based tasks.

First, it presents a 1,000-sample benchmark comprising human-curated audio–question–answer triplets sourced from real-world internet videos. Each sample includes a chain-of-thought rationale and is categorized into diverse hierarchical reasoning layers.

Second, the authors conduct a comprehensive evaluation of 30 audio-capable models on MMAR, covering a broad range of approaches, including open- and closed-source LALMs, reasoning-augmented LARMs, Omni models, and cascaded LLM/LRM pipelines.

Key insights from the evaluation show that open-source models perform only marginally better than random guessing, while closed-source and reasoning-augmented models achieve significantly higher accuracy.

Finally, ablation studies and error analysis highlight persistent challenges, particularly perceptual and reasoning errors, emphasizing the need for improved training data and architectural innovations to advance deep audio reasoning.

**Dataset Code Accessibility:**

Yes

**Dataset Code Comments:**

open source assets: https://github.com/ddlBoJack/MMAR?tab=readme-ov-file

metadata of MAR: https://github.com/ddlBoJack/MMAR/blob/main/MMAR-meta.json

**Ethical Considerations:**

No, there are no or only very minor ethics concerns

**Limitations Weaknesses:**

1. Data item bias:

In the ablation experiments, the authors observe that “a small subset of questions may still be answerable using text priors alone—indicating residual bias.” This implies that some tasks could be “gamed” by language models without truly grounding in the audio modality, a known issue in other foundation model datasets as well. It is encouraging to see that the authors propose the use of automatic bias-detection protocols, such as testing each question against a text-only baseline and removing or rewriting items that exceed a minimal text-only accuracy threshold.

2. Underrepresented sub-categories:

Certain domains, such as counterfactual reasoning or non-speech/music tasks in rare genres, are sparsely represented, leading to a long-tail distribution in the dataset. Addressing this imbalance is important. Some post-processing steps maybe helpful, such as prioritizing annotation efforts in underrepresented areas by identifying sub-categories with the lowest sample counts (as shown in Figure 2). So it is good to see that the paper outlines a clear plan for expanding MMAR in the future, especially in underrepresented sub-categories and modalities.

**Strengths Contributions:**

1. First reasoning benchmark for audio foundation models:

MMAR is the first benchmark explicitly targeting deep reasoning in the audio foundation model domain. It contains 1,000 human-curated audio–question–answer triplets, each annotated with a detailed cot rationale, making it valuable in the current era of reasoning-focused models. The dataset is organized into four hierarchical reasoning layers, Signal, Perception, Semantic, and Cultural, enabling fine-grained assessment of reasoning capabilities.

2. Clear presentation and good writing:

It is well-structured and easy to follow, with informative figures and tables that aid understanding and reproducibility. For example, Figure 1 provides illustrative task examples, Figure 2 presents data statistics, and Table 1 offers comprehensive benchmark comparisons.

3. Comprehensive model evaluation and insightful analysis:

It conducts a comprehensive evaluation of 30 audio-capable models across five model categories, indicating the superiority of reasoning-augmented and closed-source models in narrowing the performance gap.

Additionally, it presents valuable error analysis, showing that a significant portion of errors arise from misalignment between reasoning chains and final outputs, which could inform future training efforts focused on data quality and reasoning alignment.

---

> ### Author Rebuttal · Authors · 2025-07-31
>
> Dear Reviewer R18W,
>
> We sincerely thank the reviewer for their thorough reading, affirmative comments, and constructive feedback regarding our MMAR benchmark paper. We are pleased the reviewer acknowledges (i) the unique value of MMAR as the first deep audio reasoning benchmark; (ii) the clarity and comprehensiveness of the paper; (iii) the technical merit and reproducibility; and (iv) our comprehensive evaluation and insightful analysis. Below, we address the concerns in detail and outline concrete steps for improvement.
>
>
> ## W1: Data item bias.
>
> We appreciate your insightful comment regarding the observation that a small subset of our QA still can be answered based on linguistic priors. We agree that this indicates language bias exists, a known but critical issue for any multimodal benchmark, which phenomenon we have discussed in Section 6.2. Below, we detail our three-step de-biasing pipeline, which we have now integrated into our work.
>
> ### 1. Identifying the Source of Bias
>
> **We first hypothesized that the bias most likely originates from the distractors rather than the question-answer pairs themselves**. This is because our multi-stage curation process (involving expert question setter, correctors, and quality inspectors) already included rigorous checks to ensure questions required audio grounding. The automated generation of distractors, however, can inadvertently introduce linguistic patterns that powerful LLMs might exploit. Our subsequent analysis confirmed this hypothesis.
>
> ### 2. Determining Which Questions Require Mitigation
>
> We designed an automatic protocol to detect questions requiring mitigation. **The intuition is simple: if a strong text-only model can consistently answer a question correctly without the audio, the question is biased.**
>
> We operationalized this using a statistical test:
> 1. For each question, we use a powerful text-only model (gpt4.1) to answer it $n=12$ times, using only the question and multiple-choice options.
> 2. We calculate the probability of achieving the observed accuracy or higher by chance, using a binomial distribution $X \sim Binom(n,1/c)$, where $c$ is the number of choices. This yields a p-value.
> 3. **Any question with a p-value below our significance level $\alpha = 0.02$ is flagged as biased and targeted for mitigation.** This provides a robust, data-driven method to pinpoint problematic items.
>
> ### 3. Our Approach to Removing Language Bias
>
> We employed two rewriting methods to mitigate language bias caused by distractors:
>
> 1. **Method 1: Distractor-centric Generation to Break Semantic Symmetry.**:
> Refer to [1], if every negative answer choice is generated by changing a small part of the correct answer, the LLM can detect those changes to find a “centralized” description and use that cue for its prediction.
> For a given correct answer $C$, instead of generating three distractors based on $C$, we first generate a primary distractor $D_1(C)$. We then generate the remaining distractors, $D_2(D_1)$ and $D_3(D_1)$, based on the first distractor. This breaks the semantic symmetry around the correct answer, forcing the model to evaluate each option on its own merit rather than through pattern recognition.
>
> 2.  **Method 2: Adversarial Distractor Generation.**:
> Inspired by valuable suggestion from Reviewer 66d2, we introduced adversarial distractors. For a given question, we generate an option that is highly plausible based on common-sense or real-world priors but explicitly incorrect according to the audio.
> This adversarial choice is designed such that models unable to genuinely comprehend the audio content are more likely to incorrectly select it.
>
> ### 4. Experimental Validation
>
> To prove the efficacy of our de-biasing pipeline, we re-ran the "noise input" experiment on both the original and revised MMAR versions. This setting isolates the models' reliance on text priors. The results are below:
>
> |#|Exp|Method1|Method2|Qwen-2.5-Omni-7B|Qwen-2.5-Omni-3B|
> |:--:|:--:|:--:|:--:|:--:|:--:|
> |1|Random Guess (Theoretical)|✘|✘|29.3±4.8|29.3±4.8|
> |2|Original MMAR (w/ Noise)|✘|✘|36.3|35.8|
> |3|Exp1 (w/ Noise)|✔|✘|33.4|34.1|
> |4|Exp2 (w/ Noise)|✔|✔|32.6|33.9|
>
> *Table: Impact of language bias mitigation on model performance with noise input. Line 1 shows scores of the theoretical random guess, with random range with significance level of $\alpha=0.001$.*
>
> As the table shows, the performance on the original MMAR (Line 2) was slightly above the random guess baseline, confirming the presence of some language bias. Our mitigation methods (Lines 3 and 4) systematically reduce this performance. The final, de-biased MMAR (Line 4) brings the models' performance squarely within the statistical range of random guessing, which demonstrates that models are less able to exploit linguistic shortcuts.
>
> ## W2: Regarding Underrepresented Sub-categories.
> We agree with your assessment that certain advanced reasoning sub-categories, such as counterfactual reasoning, are less represented in the current version of MMAR.
> We believe the current MMAR, even with this distribution, provides a robust and critically needed foundation for the field. **The results clearly show that even on the more populated categories, current state-of-the-art models struggle significantly**, highlighting the benchmark's immediate value and challenging nature.
> Besides, we appreciate your acknowledging our plans for future expansion. To make this commitment more concrete, **we will update to outline a clear and actionable roadmap for the next version of MMAR**. This roadmap will prioritize annotation efforts in the specific underrepresented tasks.
>
> ## Summary
> Once again, we express our sincere gratitude for your review. Given the paper's pioneering contributions, which you kindly acknowledged, and the significant improvements on mitigation of language bias, we respectfully ask if you would consider raising your score to better reflect the strengthened contribution and impact of this research.
>
> ## Reference
> [1] Cai, Mu, et al. "Temporalbench: Benchmarking fine-grained temporal understanding for multimodal video models." arXiv preprint arXiv:2410.10818 (2024).

---

> > ### Author Response · Authors · 2025-08-07
> > **Acknowledgement**
> >
> > Dear Reviewer,
> >
> > We would like to let you know that the discussion phase for NeurIPS has been extended by two days. If our previous responses have addressed your concerns and resolved your doubts, we would greatly appreciate it if you could consider updating your score accordingly.
> >
> > If you have any further questions or concerns, please feel free to continue the discussion. We are more than happy to provide additional clarifications or details.
> >
> > Thank you again for your valuable feedback!
> >
> > Best regards,
> >
> > Authors

---

### Official Review · Reviewer_jjXE · 2025-07-05

**Rating:** 5
**Confidence:** 4

**Summary:**

This paper introduces MMAR, a new benchmark specifically designed to evaluate the deep reasoning capabilities of Audio-Language Models (ALMs) across a wide array of real-world audio scenarios. MMAR contains 1,000 high-quality audio–question–answer triplets, annotated with Chain-of-Thought (CoT) rationales and categorized into four hierarchical reasoning levels: Signal, Perception, Semantic, and Cultural.

**Dataset Code Accessibility:**

Yes

**Ethical Considerations:**

No, there are no or only very minor ethics concerns

**Final Justification:**

I have carefully read the rebuttal and other reviews. The response solved my concerns. I keep my positive score.

**Limitations Weaknesses:**

W1: The dataset contains only around 1,000 QA pairs, which limits its scalability and statistical reliability for training large models. The relatively small volume may also hinder a comprehensive evaluation across diverse reasoning capabilities.

W2: Although the inclusion of Chain-of-Thought (CoT) annotations is a notable strength, the quality of these rationales has not been rigorously validated. Without such validation, it is difficult to assess the true utility of these rationales for model supervision or explanation.

**Strengths Contributions:**

S1: MMAR addresses a significant gap in multimodal research by focusing on deep audio-language reasoning, a relatively underexplored yet critical area for AI understanding.

**One of the key strengths of this dataset lies in the fact that its high-level reasoning questions are human-curated rather than generated by large language models (LLMs). This distinguishes it from many existing datasets, where the reliance on LLM-generated questions may introduce biases or lack the depth and diversity found in human-authored content.**

S2: The design of challenging questions by brainstorming and heuristic-based data annotation is interesting and significant.

S3: Including CoT rationales is a good contribution, supporting research not only in answer accuracy but also in explainability and step-wise reasoning.

---

> ### Author Rebuttal · Authors · 2025-07-31
>
> Dear Reviewer jjXE,
>
> We are sincerely grateful for your thorough review, positive assessment, and insightful feedback on our paper.
> We are particularly encouraged that you (i) recognize the core contributions of our work addressing a critical gap in deep audio reasoning, (ii) highlight the value of our human-curated QA, (iii) recognize the interesting and significant approach of designing challenging questions through brainstorming and heuristic-based annotation, and (iv) appreciate the inclusion of CoT rationales.
> We will now address the two mentioned weaknesses with further clarification, hoping to fully solidify your confidence in our work.
>
> ## W1: Dataset size.
> We appreciate your concern regarding the dataset size, which touches upon a crucial aspect of our benchmark's design that we also address in our Limitations (Appendix M). Our guiding principle was **"quality over quantity"**, which is especially vital for a benchmark targeting **deep audio reasoning**. To elaborate specifically:
>
> 1. **High Cost of Annotating Deep Reasoning Tasks**: As you noted in S2, our question design process is highly intricate. **As detailed in Appendix M, finding and annotating a single data point took 10-30 minutes on average per expert, excluding the multi-stage verification and refinement process**. This effort is fundamentally different from that for shallow understanding tasks that can be generated at scale.
> 2. **High Expertise Required for Annotators**: As detailed in Appendix B, our annotation and review team was composed exclusively of domain experts, including PhDs in speech processing, audio AI researchers, and specialists from leading music conservatories. This commitment to expert-driven quality was paramount.
> 3. **Comparison with Leading Benchmarks**: Our scale is well-aligned with other highly-regarded, expert-driven benchmarks. For instance, Video-MMMU [1] contains 900 QA pairs from 300 videos, LVBench [2] has 1,549 QA pairs from 500 videos, and OmniBench [3] has 1,142 samples. In the pure reasoning domain, the AIME [4] math competition comprises only 30 problems annually. This places MMAR's scale squarely within the established norms for benchmarks where **quality and difficulty are the primary metrics of value**.
>
> As we acknowledged in our paper's Limitations section (Appendix M), *"We recognize this as a constraint on coverage and plan to expand MMAR in the future, particularly in underrepresented sub-categories and modalities."* This reflects our commitment to the long-term growth of this benchmark.
> While we are committed to its future expansion, we believe the current MMAR, as presented, already constitutes a complete and substantial contribution to the community.
>
> ## W2: Quality validation of CoT rationales.
> Thank you for raising this critical point about CoT quality, as it is central to the benchmark's utility and long-term value. **In fact, we have withheld reasoning cues and CoT annotations, which will be released at an appropriate time, to prevent potential data leakage into training for reasoning models, as at present it is still in the early stage of development in the field of audio deep reasoning.**
>
> Before we open-source the reasoning cues and the CoT annotations, **we will go through the same multi-stage process involving separate individuals for authoring, correction, and final quality inspection, as described in Sec 3.2 and Fig 3**.
>
> ## Summary
> Once again, we thank you for your time and constructive feedback. We are confident that these clarifications will address your concerns and further highlight the value of MMAR as a rigorous and impactful benchmark.
>
> ## Reference
>
> [1] Hu, Kairui, et al. "Video-MMMU: Evaluating knowledge acquisition from multi-discipline professional videos." arXiv preprint arXiv:2501.13826 (2025).
>
> [2] Wang, Weihan, et al. "LVBench: An extreme long video understanding benchmark." Proc. ICCV (2025).
>
> [3] Li, Yizhi, et al. "OmniBench: Towards the future of universal omni-language models." arXiv preprint arXiv:2409.15272 (2024).
>
> [4] https://huggingface.co/datasets/opencompass/AIME2025

---

### Decision · Program_Chairs · 2025-09-18

**Decision:**

Accept (poster)

**Comment:**

This paper introduces MMAR, the first benchmark explicitly targeting deep audio-language reasoning, comprising 1,000 human-curated audio–QA pairs with Chain-of-Thought rationales and a four-level reasoning taxonomy. Reviewers consistently highlighted the novelty, rigorous expert-driven data collection, and the benchmark’s clear contribution in exposing critical limitations of current ALMs. Strengths include the high quality of annotation, coverage of diverse real-world modalities, and comprehensive evaluation of 30 models with insightful error analysis. Main concerns centered on the relatively modest dataset size, residual language bias, and underrepresentation of some subcategories, though the rebuttal provided detailed justifications (quality over quantity, expert annotation costs) and demonstrated a principled de-biasing pipeline. With reviewers converging on positive assessments and weaknesses judged minor or addressable, the overall contribution is timely, impactful, and well-executed.